# SmallKV: Small Model Assisted Compensation of KV Cache Compression for Efficient LLM Inference

**Yi Zhao**[1][*] **Yajuan Peng**[2][*]**, Cam-Tu Nguyen**[3],
**Zuchao Li**[4]**, Xiaoliang Wang**[3]**, Hai Zhao**[1][†]**and Xiaoming Fu**[5,2][†]
[1]*AGI Institute, School of Computer Science, Shanghai Jiao Tong University*
[2]*Shanghai Key Laboratory for Intelligent Information Processing, Fudan University*
[3]*State Key Laboratory for Novel Software Technology, Nanjing University*
[4]*School of Artificial Intelligence, Wuhan University*
[5]*Institute of Computer Science, University of Göttingen, Göttingen, Germany*

## Abstract

KV cache eviction has emerged as an effective solution to alleviate resource constraints faced by LLMs in long-context scenarios. However, existing token-level eviction methods often overlook two critical aspects: (1) their irreversible eviction strategy fails to adapt to dynamic attention patterns during decoding (the saliency shift problem), and (2) they treat both marginally important tokens and truly unimportant tokens equally, despite the collective significance of marginal tokens to model performance (the marginal information over-compression problem). To address these issues, we design two compensation mechanisms based on the high similarity of attention matrices between LLMs of different scales. We propose SmallKV, a small model assisted compensation method for KV cache compression. SmallKV can maintain attention matching between different-scale LLMs to: 1) assist the larger model in perceiving globally important information of attention; and 2) use the smaller model's attention scores to approximate those of marginal tokens in the larger model. Extensive experiments on benchmarks including GSM8K, BBH, MT-Bench, and LongBench demonstrate the effectiveness of SmallKV. Moreover, efficiency evaluations show that SmallKV achieves 1.75 - 2.56 times higher throughput than baseline methods, highlighting its potential for efficient and performant LLM inference in resource constrained environments.

## 1 Introduction

Large language models (LLMs) [29] have emerged with remarkable natural language understanding capabilities and broad application prospects. Despite the advancements, the deployment of LLMs is hindered with significant computational challenges and high GPU memory consumption, particularly when processing long contexts. This issue arises from the intrinsic complexity of their self-attention mechanism, which scales quadratically with the length of the input sequence. Recent developments in reasoning models, exemplified by ChatGPT-o1 [28] and DeepSeek-R1 [7] have exacerbated this issue due to their lengthy reasoning process.

---

[*]**Equal contribution**: **Yi Zhao** (**zhao-yi@sjtu.edu.cn**) and **Yajuan Peng** (**yjpeng24@m.fudan.edu.cn**)

[†]Corresponding author: Hai Zhao (zhaohai@cs.sjtu.edu.cn) and Xiaoming Fu (fuxiaoming@fudan.edu.cn). This work was supported by The Major Program of Chinese National Foundation of Social Sciences under Grant 'The Challenge and Governance of Smart Media on News Authenticity' (No. 23ZD213), the National Natural Science Foundation of China (No. 62306216), and partially supported by the EU Horizon Europe CODECO project (Grant No. 101092696).

Numerous studies have demonstrated the high degree of sparsity within attention mechanism, leading to the development of various Key-Value cache (KV cache) compression methods such as quantization [15, 9], eviction [51, 33, 43], and merging [17, 38, 27]. These methods significantly reduce GPU memory usage and enhance the throughput of inference systems. Our work focuses on eviction-based methods, which identify and retain only the critical tokens to reduce KV cache consumption while minimizing performance degradation in a training-free manner.

Current KV cache eviction methods, however, face two key challenges: (1) the saliency shift issue caused by dynamic changes in token importance during decoding, where permanent token removal strategies become suboptimal when the decoding process evolves, and (2) the classification of tokens into critical/unimportant categories fails to account for marginal tokens that collectively contribute significantly to model performance despite their individually modest attention scores. Existing approaches lack mechanisms to adapt to shifting saliency patterns or to apply differentiated treatment to these three distinct token categories (critical, marginal, and unimportant), leading to either excessive memory consumption or unnecessary quality degradation.

A previous study [4] has revealed a notable similarity in attention patterns between small and large models within the BERT architecture. We further discover the similar observation in the decoder-only architecture models, leading to a novel perspective for addressing the aforementioned limitations. We then propose **S**mall **M**odel **A**ssisted Compensation of **KV** Cache Compression for Efficient **LLM** Inference (**SmallKV**), which introduces a small language model (SLM) to perform *saliency shift compensation* and *marginal information compensation* for KV Cache Compression (of LLM). Specifically, the saliency shift compensation mechanism leverages the SLM to maintain global critical information and help identify evicted tokens that may regain significance. Meanwhile, the marginal information compensation identifies tokens with relatively lower attention scores, yet still contribute to model performance. These tokens are less sensitive to the approximation of attention, allowing us to leverage attention scores of SLM for compensation. This results in a hierarchical compression strategy that differentiates and compresses tokens based on their varying levels of importance. It should be noted that SmallKV is compatible with efficient attention implementation such as Flash Attention, which significantly enhances the efficiency of inference. In practical deployment, SmallKV can be used with speculative decoding [18] to speed up LLM inference further.

We conduct a comprehensive evaluation of SmallKV across several benchmarks, including GSM8K [6], BBH [34], MT-bench [54], and Longbench [1]. The experimental results indicate that SmallKV consistently delivers superior performance, especially under low KV cache budgets. Experiments on different model series (e.g., the Qwen series and LLaMA series) and model sizes (ranging from 7B to 72B) highlight the robustness and generalizability of our approach. Efficiency evaluations show that SmallKV achieves 1.75 - 2.56 times higher throughput compared to previous KV cache compression methods. These experimental results collectively offer compelling evidence of SmallKV's accuracy and efficiency as a compensation plugin for KV cache compression.

## 2   Related Work

The approaches to KV cache compression can be broadly categorized into three classes: eviction, merging, and quantization. Eviction [30, 31, 19, 25, 55, 49] aims to retain only a small set of critical tokens' KV caches to achieve nearly lossless inference performance. Merging leverages the high angular similarity observed among deep KV caches to reduce both intra-layer [17, 38, 27, 50, 37, 39] and cross-layer [45, 23] redundancies. Quantization compresses the data by mapping the original full-precision tensor values to discrete levels and storing them at lower precision, including model weight quantization [33, 46, 32] and KV-cache-only quantization [15, 9, 44, 8, 16].

Numerous eviction methods have been proposed, which can be categorized into static and dynamic strategies. **Static strategies** [10, 21, 47] perform token filtering during prefilling and maintain KV cache of fixed size throughout the subsequent decoding steps (e.g., sliding window attention [2]). A further approach is to maintain both the initial and the recent tokens [41, 12]. **Dynamic strategies** [51, 43, 52, 5] continuously update the critical KV cache during decoding, while the KV cache of unimportant tokens will be permanently removed or offloaded from the GPU. The core of **permanent eviction** lies in selecting critical tokens to minimize the damage to model accuracy. $H_2O$ [51] observed that almost all layers exhibit a sparsity exceeding 95%, indicating that maintaining just 5% of the KV cache based on the accumulative attention scores of each token is sufficient for decoding the

same output token at each generation step. PyramidInfer [43] employs a pyramid-shaped hierarchical processing approach, where recent tokens are assigned greater weight and the length of KV caches in deeper layers is reduced. Permanent token removal has two main limitations. Irreversible token eviction can degrade model performance in multi-turn dialogue scenarios and long-sequence tasks. Additionally, relying on attention scores prevents the model from adapting to some acceleration techniques (e.g., Flash Attention). Consequently, **non-permanent eviction** [40, 35, 48, 14, 24] has been proposed by dynamic cache offloading based on indexing. These methods manage and access the KV cache offloaded to multi-tier cache systems at the granularity of chunks or clusters. However, achieving fast and accurate retrieval with high precision remains challenging, while the index construction for retrieval introduces significant decoding latency.

Speculative decoding [20, 18, 3, 53] is a widespread technology to accelerate inference. It employs a pair of models, where the smaller one generates candidate tokens in advanced and the larger one verifies them in parallel. By enabling parallel token generation, speculative decoding addresses the inefficiencies inherent in traditional autoregressive approaches, achieving speedups of over three times without compromising the quality of the generated output. This technique has been widely adopted in commercial LLMs [3, 22].

## 3 Pilot Observation

### 3.1 Saliency Shift Issue

The dynamic nature of LLM decoding leads to the **shift in token saliency**, which is the change of the token set with high attention over time. Most existing KV cache eviction strategies, which rely on **permanent token removal** [51, 43], are highly susceptible to this issue. To address this, we leverage a **small language model (SLM)** from the same model series to approximate the LLM's attention scores. This approach aligns well with speculative decoding [20, 18, 3, 53, 22], where the SLM generates candidate tokens in advance and the LLM verifies them in parallel, thereby allowing the combination of these two appealing approaches for further optimization of inference speed.

**Observation 1.** The saliency shift leads to the discrepancy of the important tokens between the compressed KV cache view and the uncompressed global view, and this gap causes the model to lose critical information during inference. We mimic the continual process of KV cache compression in Figure 1 (a). Here, in "real-drop" procedure, we perform two times compression, mimicking the continual compression in real decoding scenario. In the first compression, the important token (yellow) information in the first half of the dashed line is retained, while the remaining unimportant tokens (blue) are evicted and their KV cache are permanently removed. The important tokens, along with the new ones in the second half of the dotted line, continue to participate in subsequent decoding and the second-time compression. However, at the second time compression, the global view of an uncompressed cache shows differences in the selection of important tokens (red), i.e., the shift in token saliency between the second compression step and the global view-based compression.

Specifically, we quantified the gap caused by saliency shift on the Wikitext-v2 [26] dataset using Qwen2-7B in Figure 1 (b). The importance of tokens is measured by accumulative attention scores, following $H_2O$[51]. We measure the discrepancy between sets of important tokens derived from the real drop view and the global view using Jaccard similarity. The relatively low Jaccard similarity values, ranging from 0.55 to 0.77, indicate a significant difference between realistic KV cache eviction methods and the global ground truth, which means a considerable number of globally important tokens are erroneously evicted under the real drop method. In addition, the similarity maintains a clear declining trend as the KV cache budget decreases, indicating that the influence brought by saliency shift is more serious. This is particularly concerning since most current KV cache compression methods claim to operate with a low KV cache budget (e.g., 10%–30%). Therefore, an effective approach to compensate for saliency shift issue is critical and necessary.

**Insights 1.** The core issue with the saliency shift lies in the inability of existing KV cache compression methods to effectively maintain past information after compression. These methods predominantly focus on LLM itself without considering external information. We, however, observe that LLMs of different sizes within the same series tend to exhibit highly consistent attention patterns. Therefore, we propose a collaborative approach during inference, where an assisted SLM works with the LLM in parallel, using the attention matrices of the SLM for saliency shift compensation.

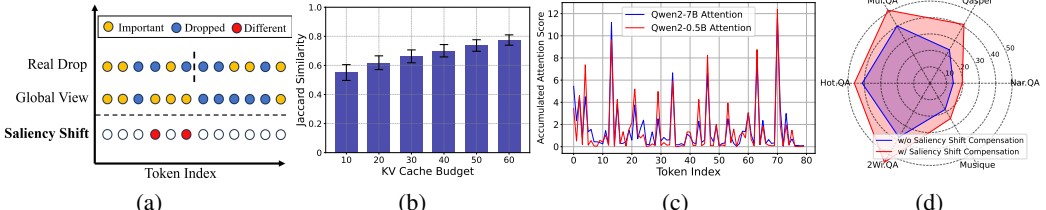

(a)          (b)          (c)          (d)

Figure 1: (a) The illustration of saliency shift. (b) The quantification of saliency shift issue by measuring Jaccard similarity of important tokens between real drop and global view of critical tokens. Lower Jaccard similarity indicates that more important tokens have been wrongly evicted. (c) The visualization of highly consistent attention patterns between LLMs with different scaling, indicating that SLM can assist in capturing and preserving global attention information. (d) Comparison of 10% KV cache budget for saliency shift compensation using Qwen2-0.5B to assist Qwen2-7B.

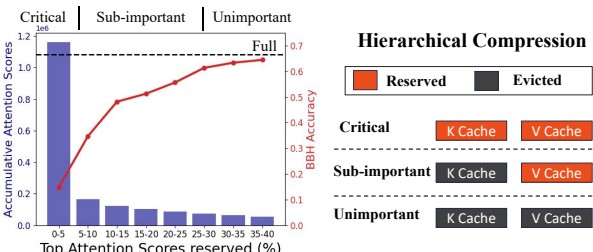

Figure 2: Top: The distribution of attention sparsity (blue bar) along with the drop of model accuracy (red line), indicating the necessity of maintaining marginal tokens. Horizontal line (Full) represents the baseline of full KV cache. Bottom: Marginal information compensation by the hierarchical compression that differentiates tokens based on their varying levels of importance.

We measure the similarity of the attention patterns between Qwen2-0.5B and Qwen2-7B on Wikitext-v2 [26] dataset. Figure 1 (c) illustrates one example of the trends of accumulative attention scores between SLM and LLM, as more samples and detailed discussion in Appendix B. For each attention head in LLM, we search the most similar one in the SLM for matching. The quantitative analysis shows that average cosine similarity between Qwen2-0.5B and Qwen2-7B after similarity matching reaches 0.947. The result also visualizes a high degree of consistency, particularly among tokens with high attention scores. This finding indicates a promising way to effectively address the saliency shift issue without introducing significant overhead. We evaluate the effectiveness of saliency shift compensation with 10% KV cache budget across six downstream tasks, as shown in Figure 1 (d). The results demonstrate that using Qwen2-0.5B as the assisted SLM for saliency shift compensation significantly outperforms using only Qwen2-7B without any compensation.

### 3.2 Marginal Information Overcompression

Current KV cache eviction methods divide tokens into two categories—critical and unimportant—overlooking the presence of **marginal tokens**. These tokens, while less impactful than critical ones, still hold significant relevance and collectively contribute far more to model performance than their attention score suggests. Unlike unimportant tokens, marginal tokens are essential for preserving output quality, yet they are currently subjected to the same aggressive compression or eviction strategies as truly negligible tokens. To address this, a dedicated approach is needed to: (1) **distinguish** marginal tokens from unimportant ones, and (2) **apply tailored compression strategies** (rather than outright removal) to retain their contribution.

**Observation 2.** We measure the accumulative attention scores of each token across all heads of Qwen2.5-14B on the BBH dataset [34], and statistically rank these scores in Figure 2 to reveal the necessity of maintaining marginal tokens. Consistent with previous research findings, we also observe that only a small fraction of tokens account for the majority of attention scores. The sum

of the accumulative attention score for the top 5% of tokens is approximately 7 times greater than that of the 5%-10% range, while the 5%-10% range is only 1.34 times higher than the 10%-15% range, highlighting the dominant position of the top 5% of tokens in attention scores. However, the compression of tokens in range of 5%-15% has caused the accuracy to drop from 0.482 to 0.148 under $H_2O$ method.

**Insights 2.** Previous methods, however, fail to recognize the necessity of maintaining marginal tokens, and conflate them with critical tokens or unimportant tokens. This leads to an underutilization of attention sparsity, inducing a mandatory trade-off between KV cache consumption and model performance degradation. Our observations suggest that maintaining marginal tokens is necessary and that they can tolerate approximation compared to critical tokens (extended analysis can be referred to Appendix C). Therefore, a reasonable approach is to apply different compression strategies for tokens of varying importance levels, as shown in Figure 2. Critical tokens retain the full KV cache to avoid precision loss, marginal tokens retain only the V cache and use SLM to approximate the attention mechanism, and unimportant tokens are evicted completely.

As insight 1 indicates that LLMs of different sizes within the same series exhibit highly consistent attention patterns, we propose compensating for marginal tokens based on the attention scores from the SLM. Specifically, we approximate the attention mechanism of marginal tokens by multiplying their V cache with the corresponding attention scores derived from the SLM. This method reduces the K cache consumption for these tokens while maintaining their contribution to overall model performance. By doing so, we effectively bridge the gap between theoretical attention sparsity and practical performance, thereby enhancing the efficiency and efficacy of KV cache compression.

## 4   SmallKV Method

We introduce the insight of SmallKV in Section 3, involving saliency shift compensation and marginal tokens compensation assisted by the SLM. In this section, we provide a detailed description.

**Similarity Matching.** In prefill stage, SmallKV establishes the similarity matching between SLM and LLM. Given an LLM $M$ and the corresponding SLM $M_s$, the prompt is first forward in both models to obtain all attention matrices respectively. The attention matrices of M are denoted as $A_i = Softmax\left(\frac{Q_i K_i^\top}{\sqrt{d_h}}\right) \in \mathbb{R}^{n \times n}, 0 \leq i < L \cdot D$, and those in $M_s$ are denoted as $A'_j$, $0 \leq j < l \cdot d$. $Q \in \mathbb{R}^{n \times d}$ and $K \in \mathbb{R}^{n \times d}$ denote query matrix and key matrix in attention mechanism. $L$ and $l$ represent the number of layers in the $M$ and $M_s$. $D$ and $d$ denote the number of attention heads per layer in each model. $n$ represents the number of tokens in the prompt. Due to differences in model scaling, we have $l \cdot d << L \cdot D$. Let $C$ be the KV cache of the context, the accumulative attention score vector in a specific context can be expressed as:

$$F(A_i, C) = (s_i^1, s_i^2, \ldots, s_i^n), \text{ where } s_i^v = \sum_{u=1}^{n} A_i[u, v] \tag{1}$$

We then can calculate pairwise similarity between $A_i$ and $A'_j$ according to Jaccard simularity of their TopK indices as:

$$S(A_i, A'_j) = \frac{\left|TopK\left(F(A_i, C)\right) \cap TopK\left(F(A'_j, C)\right)\right|}{\left|TopK\left(F(A_i, C)\right) \cup TopK\left(F(A'_j, C)\right)\right|} \tag{2}$$

The mapping function that maps $i$-th attention matrix of LLM to the $j$-th one in SLM is obtained by:

$$f(i) = argmax_j S(A_i, A'_j) \tag{3}$$

**Saliency Shift Compensation.** Previous eviction strategies remove tokens based on the policy:

$$\mathcal{E}(A_i, C) = C \setminus \{v\}, \text{ where } \{v\} \leftarrow argmax_{\{v\} \in C} F(A_i, C) \tag{4}$$

The saliency shift problem in the previous methods can be formally defined as:

$$\mathcal{E}(A_i, C_{all} \cup \{t\}) \neq \mathcal{E}(A_i, C_r \cup \{t\}) \tag{5}$$

where $C_{all}$ denotes the full KV cache, $C_r$ represents the compressed cache obtained after prefilling, and $t$ indicates the recently added token. The equation suggests that the selection of critical tokens from a compressed cache (e.g., after prefilling) is different from that if we have the full cache.

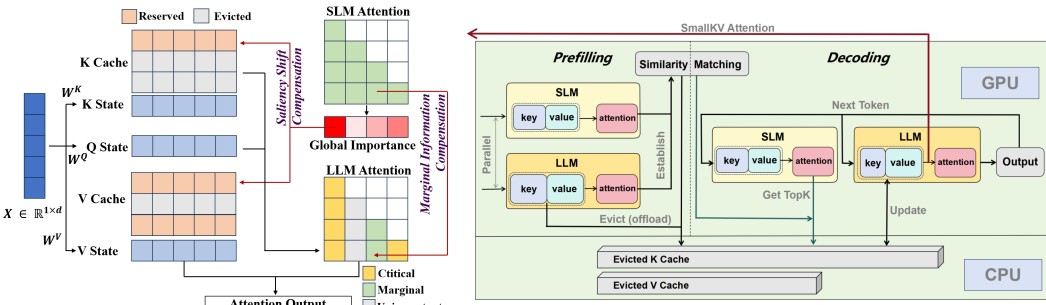

Figure 3: The attention illustration of the SmallKV method in decoding.

Figure 4: The system architecture of SmallKV. (The detail of SmallKV attention compensation is in Figure 3)

Unlike previous methods, SmallKV maintains the full cache of SLM $C_{all}^s$, and performs eviction for the $i$ cache of LLM based on he $f(i)$-th (full) cache of SLM and the eviction policy $\mathcal{E}(A'_{f(i)}, C_{all}^s)$.

**Marginal Information Compensation.** In decoding stage, let the attention be $A_i \in \mathbb{R}^{1 \times n}$, the vanilla attention can be expressed as $O_i = A_i \cdot V_i \in \mathbb{R}^{1 \times d}$, where $V_i \in \mathbb{R}^{n \times d}$ denotes the value matrix. The attention mechanism for Marginal Information Compensation can be expressed as $O_i^* = A_i^* \cdot V_i$, where $A_i^*$ is calculated as follows:

$$
A^*[k] = \begin{cases} A_i[k], & \text{if } k \in TopK\left(F(A'_{f(i)}, C_{all}^s)\right), \\ A'_{f(i)}[k], & \text{if } k \in Top(P-K)\left(F(A'_{f(i)}, C_{all}^s)\right), \\ 0, & \text{otherwise.} \end{cases}
\tag{6}
$$

where $K$ and $P - K$ denote the numbers of critical tokens and marginal tokens, which are determined by KV cache budget.

The formula shows that for marginal tokens that appear in the $Top(P-K)$ importance, we approximate their attention scores using the corresponding elements from SLM $M_s$ to avoid using K cache. This approach allows for retaining important information while evicting the K cache of these tokens, effectively implementing a hierarchical compression policy that differentiates between various levels of token importance. The illustration of attention mechanism in SmallKV is shown in Figure 3.

**Method Details.** In practice, to ensure the stability of matching correspondence between SLM and LLM, we dynamically determine the timing of similarity matching based on context length. Specifically, we control the token length for similarity matching within the range of 100 to 200. This is to avoid the matching inaccuracy caused by excessively short token sequences and the distortion of similarity in high-dimensional space caused by overly long token sequences. If the current context length does not meet the minimum threshold, SmallKV delays the timing of similarity matching and KV cache eviction until the requirements are satisfied. Similarly, for context lengths exceeding the threshold, we truncate the prefill token length for similarity calculation.

The detailed procedure of SmallKV can be referred to Algorithm 1. It is important to note that SmallKV does not delete the KV cache but migrates it between GPU HBM and CPU memory, as the process of updating KV cache of critical tokens and V cache of marginal tokens in line 9 of the Algorithm. This migration process executes in parallel with the forward of LLM to prefetch KV cache for latency reduction. Within the attention of LLM, the computation of critical tokens and marginal tokens is also performed in parallel. The computation for critical tokens benefits from acceleration via Flash Attention, whereas the computation for marginal tokens involves only a matrix multiplication. The system architecture of SmallKV is shown in Figure 4.

**Algorithm 1** SmallKV Algorithm
___
**Input**: Large language model $M$; Assisted small language model $M_s$; Input $\boldsymbol{x}$; KV cache budget $\tau$;
 1: **if** Prefill **then**
 2:     Allocate KV cache budget $\tau$ to critical budget $\tau_c$, marginal budget $\tau_s$
 3:     Parallel forward process of $M(\boldsymbol{x})$ and $M_s(\boldsymbol{x})$ to get attention matrices $A_i$ and $A'_j$
 4:     Establish correspondence of $A_i$ and $A'_j$ by similarity matching (Eq. (2) and Eq. (3))
 5:     Output new token $t_i$ by $M(\boldsymbol{x})$ and append it to $\boldsymbol{t}$
 6: **else**
 7:     Forward process of $M_s(\boldsymbol{x} + \boldsymbol{t})$ to update $A'_j$
 8:     In parallel:
 9:     - Update KV cache of critical tokens and Update V cache of marginal tokens by $\mathcal{E}(A'_{f(i)}, C^s_{all})$
10:     **for** Attention layer $l \in$ layers of LLM **do**
11:         In parallel:
12:         - Calculate attention output $O_c$ of critical tokens using Flash Attention
13:         - Calculate attention output $O_m$ of marginal tokens using Eq. (6)
14:         Attention Output by $O_c + O_m$
15:     **end for**
16:     Output new token $t_i$ by $M(\boldsymbol{x} + \boldsymbol{t})$ and append it to $\boldsymbol{t}$
17: **end if**
**Output**: Output token sequences $\boldsymbol{t} = \{t_i\}_{i=1}^L$
___

# 5 Experiments

## 5.1 Experiments Setup

**Datasets.** We comprehensively evaluate the effectiveness of SmallKV in four kinds of scenarios: 1) GSM8K [6] for mathematical reasoning, 2) BBH [34] for language understanding, 3) MT-Bench [54] for multi-turn conversation, and 4) Longbench [1] for long-context scenario.

**Baselines.** We take two effective KV cache eviction methods as baselines. These methods select critical tokens according to their attention scores, which is the same as SmallKV.

$H_2O$ [51]: employs the accumulative attention score as the metric to evict unimportant KV cache.

PyramidInfer [43]: identifies that deeper layers exhibit greater redundancy and applies differentiated KV cache budgets between layers.

**Models.** Our experiments are conducted on the Qwen and LLaMA series of LLMs with different scales. Specifically, in the benchmark results (Section 5.2), we consider four pairwise combinations between the SLM and the LLM. These combinations include: Qwen2-0.5B with Qwen2-7B, Qwen2.5-0.5B with Qwen2.5-14B, Qwen2-7B with Qwen2-72B, LLaMA 3.2-1B with LLaMA 3.1-8B. The Qwen2-72B model is quantized to INT4 data type for efficient computing. The other experiments are primarily performed on the combination of Qwen2-0.5B with Qwen2-7B.

**Implementation Details.** All experiments are conducted on 8 NVIDIA A100 (80GB) GPUs. The configurations of environment include: CUDA (12.0), PyTorch (2.4.0), and huggingFace's Transformers[3] (4.45.1). We use greedy decoding to ensure the stability of the experimental results.

Consistent with previous study, we also allocate a proportion of the KV cache budget to recent tokens. We adopt a fixed ratio of 2:1:2 for allocating resources among critical tokens, recent tokens, and marginal tokens, respectively. For instance, with 20% KV cache budget, we allocate 10%, 5% and 10% budget to critical tokens, recent tokens, and marginal tokens (as marginal tokens only use V cache of LLM, it actually consumes half of budget). When the available KV cache budget becomes limited so that it cannot fully accommodate all critical tokens (e.g. 5%), we proportionally reduce the allocations for recent and marginal tokens to maintain model performance.

___
[3]https://github.com/huggingface/transformers

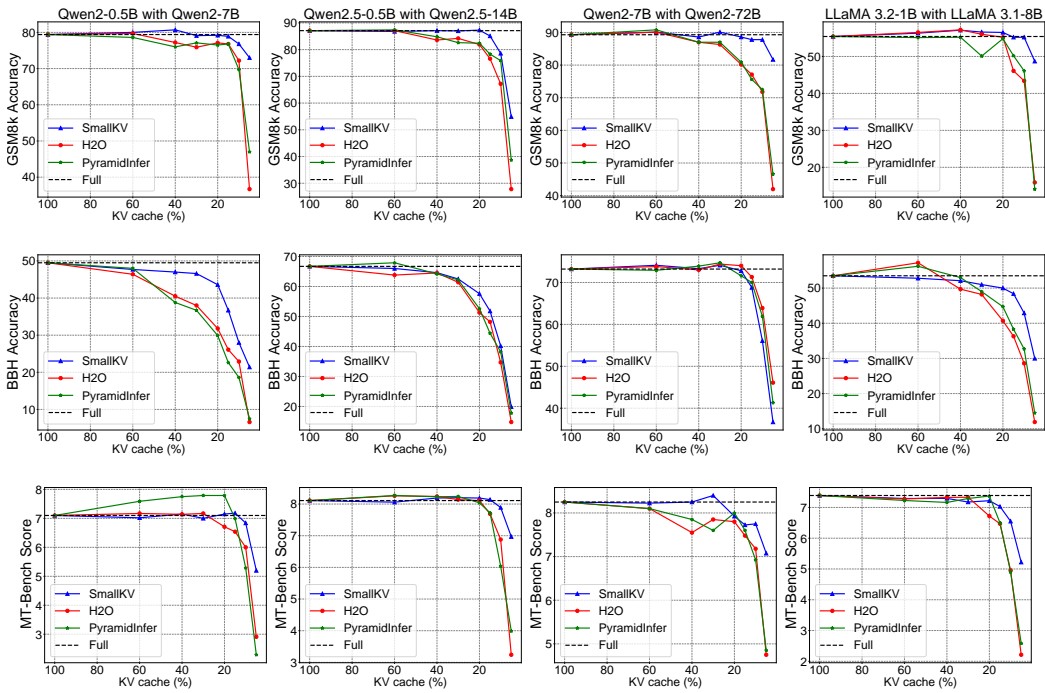

Figure 5: Benchmark results of SmallKV with KV cache budget varying from 100% to 5%. Full represents using the full KV cache without compression.

| Methods | LongBench | | | | | | | | | | | | | | | | | |
|---|---|---|---|---|---|---|---|---|---|---|---|---|---|---|---|---|---|---|
| | Single-Doc QA | | | | Multi-Doc QA | | | | Summarization | | | | Few-shot Learning | | | | Code Completion | |
| | Nar.QA | Qasper | Mul.QA | AVG | Hot.QA | 2Wi.QA | Musique | AVG | QMSum | M.News | GovRe. | AVG | Tri.QA | TREC | SAMSum | AVG | LCC | Repo. | AVG |
| *KV cache budget τ = 0.3* | | | | | | | | | | | | | | | | | | |
| SmallKV | 20.81 | 42.53 | 48.82 | **37.39** | 46.71 | 51.51 | 23.76 | **40.66** | 23.15 | 23.55 | 19.55 | **22.08** | 86.42 | 71 | 48.31 | **68.58** | 57.79 | 55.87 | 56.83 |
| H2O | 22.14 | 33.11 | 45.17 | 33.47 | 44.14 | 48.44 | 20.85 | 37.81 | 21.91 | 23.51 | 19.98 | 21.35 | 85.22 | 73 | 45.27 | 67.67 | 58.62 | 55.04 | 56.83 |
| Pyramid. | 22.36 | 32.19 | 45.14 | 33.23 | 44.14 | 46.01 | 20.77 | 36.97 | 21.33 | 22.97 | 20.08 | 21.46 | 85.42 | 72 | 46.03 | 67.82 | 58.58 | 55.67 | **57.13** |
| *KV cache budget τ = 0.1* | | | | | | | | | | | | | | | | | | |
| SmallKV | 19.85 | 42.12 | 49.63 | **37.20** | 47.74 | 51.51 | 23.26 | **40.84** | 22.84 | 22.92 | 18.90 | **21.55** | 86.49 | 71 | 47.68 | **68.39** | 58.77 | 55.14 | **56.96** |
| H2O | 14.13 | 23.50 | 40.22 | 25.95 | 41.25 | 37.89 | 18.38 | 32.51 | 20.50 | 23.22 | 20.34 | 21.35 | 84.44 | 51 | 43.69 | 59.71 | 41.33 | 44.90 | 43.12 |
| Pyramid. | 14.12 | 25.78 | 39.50 | 26.47 | 41.52 | 39.63 | 16.14 | 32.43 | 20.03 | 23.15 | 20.38 | 21.19 | 85.04 | 50 | 42.66 | 59.23 | 41.28 | 46.40 | 43.84 |
| *KV cache budget τ = 0.05* | | | | | | | | | | | | | | | | | | |
| SmallKV | 18.48 | 40.78 | 40.16 | **33.14** | 45.39 | 47.31 | 19.92 | **37.54** | 22.35 | 19.47 | 18.70 | **20.17** | 86.99 | 71 | 47.04 | **68.34** | 53.67 | 53.40 | **53.54** |
| H2O | 12.85 | 14.36 | 31.00 | 19.40 | 33.23 | 27.98 | 11.22 | 24.14 | 19.18 | 20.04 | 19.58 | 19.60 | 59.54 | 30 | 34.94 | 41.49 | 31.44 | 35.52 | 33.48 |
| Pyramid. | 13.79 | 11.43 | 30.06 | 18.43 | 36.42 | 26.67 | 9.60 | 24.23 | 19.14 | 20.36 | 19.76 | 19.75 | 61.86 | 33 | 35.43 | 43.43 | 28.24 | 29.02 | 28.63 |
| Full | 24.26 | 43.48 | 49.39 | 39.04 | 44.57 | 52.76 | 25.02 | 40.78 | 23.88 | 22.98 | 20.08 | 22.31 | 87.19 | 78 | 45.03 | 70.07 | 58.26 | 55.64 | 56.95 |

Table 1: Performance of three methods on the LongBench with different KV cache budget (using Qwen2-0.5B with Qwen2-7B). The results of full KV cache are shown at the bottom for comparision.

## 5.2 Benchmark Results

Figure 5 shows the performance of four models across a range of KV cache budgets from 100% to 5% on GSM8K, BBH, and MT-Bench. The results indicate that the SmallKV method consistently outperforms baseline approaches across nearly all models and various KV cache budgets, particularly at very low KV cache budgets. The marginal information compensation mechanism enables SmallKV to maintain attention information for a significant proportion of marginal tokens while using few KV cache, thereby sustaining high performance even at low cache budgets. For instance, in the experiment of Qwen2-0.5B paired with Qwen2-7B, with 5% KV cache budget, the scores for H2O and PyramidInfer decline from 79.4 (at full cache) to 36.7 and 47.0, respectively, whereas the SmallKV method maintains the performance score of 73.0.

Table 1 provides a comparative analysis of SmallKV method against baselines on LongBench at three distinct KV cache budgets ($τ = 0.3$, $τ = 0.1$, and $τ = 0.05$). Our method demonstrates superior performance across all five subtasks, underscoring the efficacy of SmallKV in long-context scenarios.

| Method | Bsz | Lenth | TPOT (ms) | TTFT (s) | Thr.(tokens/) |
|---|---|---|---|---|---|
| Accelerate (Eager) | | | 59.4 | 22.15 | 62.27 (1.00x) |
| Accelerate (Flash) | 64 | 2048+256 | 44.6 | 1.09 | 188.23 (3.02x) |
| $H_2O$ (20%) | | | 30.5 | 22.55 | 76.32 (1.23x) |
| SmallKV (20%) | | | 36.3 | 2.71 | **195.50 (3.14x)** |
| Accelerate (Eager) | | | 42.7 | 14.28 | 474.21 (1.00x) |
| Accelerate (Flash) | 4 | 16384+512 | 31.3 | 1.41 | 990.39 (2.09x) |
| $H_2O$ (20%) | | | 20.4 | 14.32 | 689.07 (1.45x) |
| SmallKV (20%) | | | 24.2 | 1.94 | **1203.42 (2.54x)** |

Table 2: Evaluation on efficiency of SmallKV. **Bsz**: batch size. **Lenth**: prefill length + decode lenth. **TPOT**: average time per output token in decode stage. **TTFT**: time to first token. **Thr.**: End-to-end throughput.

Similar to previous observations, SmallKV performs well with low KV cache budgets. For example, in subtasks such as Multi-Doc QA, Few-shot Learning, and Code Completion, baseline methods exhibit performance degradation with 5% KV cache budget, while the SmallKV method maintains performance comparable to that achieved with full KV cache. This highlights SmallKV's ability to effectively manage cache resources and deliver robust performance even under constrained conditions.

## 5.3 Efficiency Results

**Setup.** We conduct an efficiency analysis on one NVIDIA A100 GPU. The evaluated model is Qwen2-7B and the SLM used in SmallKV method is Qwen2-0.5B. We use synthetic datasets for testing where all prompts are padded to the same length in the batch and the outputs are also restricted to a fixed length. We consider two common scenarios: 1) Multi-user concurrency: In this scenario, the context is set to a prefill length of 2048 tokens and a decode length of 256 tokens, with a batch size of 64. 2) Extremely long context: the context is set to a prefill length of 16384 tokens and a decode length of 512 tokens, with a batch size of 4. The evaluation metric consists of average time (ms) per output token in decode stage (TPOT), time (s) to first token (TTFT), and end-to-end throughput (token/s). We use Hugging Face Accelerate and $H_2O$ as baselines. Hugging Face Accelerate employs two attention implementation methods: eager attention and PyTorch SDPA with Flash Attention kernel [11]. $H_2O$ and SmallKV both employ a KV cache budget of 20%. The times reported in SmallKV include the overhead of the assisted SLM.

**Results.** The efficiency results are shown in Table 2. It can be observed that compared to Accelerate, KV cache compression techniques ($H_2O$ and SmallKV) significantly reduce the TPOT during the decoding. This is attributed to their reduction in loading KV cache, thereby alleviating pressure on GPU memory bandwidth and decreasing the computation of attention. Specifically, introducing an additional assisted SLM in SmallKV results in increased latency during the decoding when compared to $H_2O$. However, due to its compatibility with memory-efficient attention methods, SmallKV's TTFT is notably lower than that of $H_2O$, despite the overhead incurred by similarity matching during the prefill stage. In summary, although the SmallKV method incorporates overhead such as assisted SLM and similarity matching, it benefits both from the advantages of KV cache compression and compatibility with efficient attention methods. These features collectively contribute to its superior efficiency, demonstrating significantly higher throughput than baseline methods across both scenarios.

## 5.4 Ablation Study

We conduct ablation studies on the BBH benchmark to evaluate the contributions of the saliency shift compensation and marginal information compensation components proposed in the SmallKV. The experimental results are shown in Figure 6. Firstly, it is evident that the SmallKV method exhibits a notable performance drop in the 10% to 40% KV cache budget range when the marginal information compensation component is absent. This observation aligns with the design purpose of marginal information compensation, which aims to compensate for marginal tokens beyond the small set of critical ones. Additionally, when the KV cache budget is under capacity to cover the critical tokens, the hierarchical compression for marginal tokens fails due to insufficient KV cache budget allocation for those tokens. This is reflected in the performance convergence between the w/o marginal information compensation and the full SmallKV with KV cache budgets below 10%.

If we further remove the saliency shift compensation component from the w/o marginal information compensation setup, as depicted by the w/o both compensation in the figure, there is an additional decline in performance. This highlights the effectiveness of saliency shift compensation, which

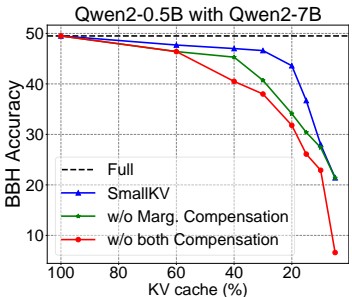
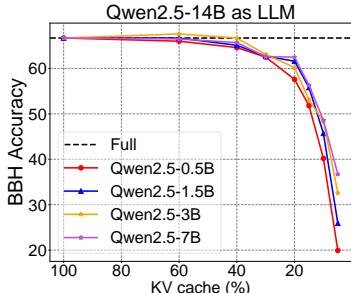

Figure 6: Ablation Study of SmallKV method.    Figure 7: The impact of SLM scaling on SmallKV.

mitigates saliency shift issue by utilizing global information from the assisted SLM. This compensation mechanism dynamically depends on context information during generation, thereby enhancing overall performance.

### 5.5   Impact of SLM Scaling

We also investigate the impact of scaling SLM on the performance of the SmallKV method. Theoretically, as the size of the SLM increases, it can provide more attention matrices, allowing the LLM to match more similar attention matrices for compensation, thereby enhancing the performance of the SmallKV method. Using Qwen2.5-14B as the LLM, we experiment with four differently scaled SLMs: Qwen2.5-0.5B, Qwen2.5-1.5B, Qwen2.5-3B, and Qwen2.5-7B on the BBH benchmark. The results are presented in Figure 7. Consistent with theoretical expectations, the performance improved gradually as the size of the SLM increased. On average, using Qwen2.5-1.5B, Qwen2.5-3B, and Qwen2.5-7B as SLMs leads to performance improvements of 4.9%, 6.8%, and 8.5% compared to using Qwen2.5-0.5B as SLM.

While the performance improvements are relatively small compared to the increase in parameters of the SLM, we also note that the influence of SLM scaling becomes more pronounced with low KV cache budgets. For instance, with 5% KV cache budget, using Qwen2.5-1.5B, Qwen2.5-3B, and Qwen2.5-7B as SLMs resulted in performance improvements of 30.2%, 63.8%, and 84.4%, respectively, compared to using Qwen2.5-0.5B. This finding indicates that under extremely constrained KV cache budgets, it is crucial to appropriately scale up the SLM to maintain performance levels. In summary, the SmallKV method necessitates a careful trade-off between the overhead introduced by scaling the SLM and the overall performance enhancement in practice.

## 6   Conclusion

We present SmallKV, a novel small-model-assisted compensation framework that addresses two critical limitations in current KV cache eviction methods: the saliency shift problem and the marginal information over-compression issue. By leveraging the high similarity of attention matrices across different model scales, SmallKV introduces two key innovations: (1) preserving globally important attention patterns through SLM assistance, and (2) accurately approximating marginal tokens using the smaller model's attention scores. Our comprehensive evaluation across multiple benchmarks (GSM8K, BBH, MT-Bench, and LongBench) demonstrates that SmallKV consistently maintains model performance even under aggressive KV cache budgets. Notably, SmallKV achieves 1.75×-2.56× throughput improvement over existing methods while remaining compatible with memory-efficient attention implementations like Flash Attention.

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

# A Overhead Analysis

In this section, we analyze the additional overhead introduced by the SmallKV method, which mainly includes the cost of the SLM and the overhead associated with the SmallKV-specific processes. It is important to note that although we provide a comprehensive analysis of these overheads, the cost related to the SLM can be **shared with the SLM used for speculative decoding in practice**, which will significantly reduce the actual additional overhead.

## A.1 Memory Overhead

The overhead of GPU memory in SmallKV comes mainly from the assisted SLM and the KV cache it stores. The storage requirement for SLM parameters is fixed and depends on the size of SLM. For KV cache consumption, it can be formulated as:

$$M_{kv} \approx 2 \cdot L \cdot N_{kv} \cdot D_{kv} \cdot S \cdot B \cdot C_b \tag{7}$$

Where $M_{kv}$ is the total GPU memory usage (in bytes) for storing KV caches. $L$ represents the number of layers in the model. $N_{kv}$ denotes the number of key-value heads used in the attention mechanism. $D_{kv}$ refers to the dimension size of each KV entry. $S$ is the sequence length, indicating the number of tokens in the sequence. $B$ stands for the batch size, which is the number of sequences processed simultaneously. $C_b$ is the number of bytes per cached value, depending on the data type used.

Here, we use Qwen2-7B as the SLM and Qwen2-72B as the LLM for illustrative analysis. The parameters for both models are outlined in the Table 3. Specifically, $N_{att}$ denotes the number of attention heads, and $D_h$ represents the hidden state dimension. Given these parameters and applying the Equation 7, we can calculate that under the same batch size and sequence length, the KV cache size of Qwen2-72B is approximately 5.71 times that of Qwen2-7B. Therefore, if the KV cache budget for the LLM (Qwen2-72B) is set to 20%, the additional overhead introduced by the SLM (Qwen2-7B) would consume about 21.9% of the KV cache saved by reducing the LLM's KV cache consumption.

It is also important to note that, in the analysis presented here, no optimizations have been applied to the SLM's KV cache. In practice, the SLM's KV cache can also be optimized through various methods, such as quantization and eviction. Implementing these optimizations can further reduce the KV cache overhead associated with the SLM.

Table 3: The config parameters of Qwen2-7B and Qwen2-72B.

|           | Qwen2-7B | Qwen2-72B |
|-----------|----------|-----------|
| $L$       | 28       | 80        |
| $N_{kv}$  | 4        | 8         |
| $N_{att}$ | 28       | 64        |
| $D_h$     | 3584     | 8192      |
| $D_{kv}$  | 128      | 128       |

## A.2 Computational overhead

The additional computational overhead in SmallKV can be divided into two main parts: forward process of the SLM and similarity matching computation between SLM and LLM.

For forward process of the SLM, it affects both prefill and decode stage. For an LLM with $L$ decoder layers, the total computational cost for one batch during a forward pass is:

$$C_{model} = L(24BSD_h^2 + 4BS^2D_h) + 2BLD_h|V| \approx 24BSLD_h^2 \tag{8}$$

Where $|V|$ denotes vocabulary size. It also can be calculated that under the same batch size and sequence length, the computational cost of LLM (Qwen2-72B) is approximately 14.9 times that of SLM (Qwen2-7B) referring to Table 3.

For the similarity matching computation between SLM and LLM, it only affects prefill stage. Assume that $S_{sim}$ denotes the sequence length for calculate similarity (typically 100-200 in SmallKV), we have the computational cost of the similarity matching for model $a$ and model $b$ as:

$$C_{sim} = (L^a N_{att}^a + L^b N_{att}^b)BS_{sim}^2 + L^a N_{att}^a L^b N_{att}^b B^2 S_{sim}^2 \approx L^a N_{att}^a L^b N_{att}^b B^2 S_{sim}^2 \qquad (9)$$

Where $L^a$ and $L^b$ denote the number of layers in the model $a$ and model $b$. $N_{att}^a$ and $N_{att}^b$ denote the number of attention heads respectively.

### A.3 Quantitative Analysis and Optimization of SLM

We quantify the theoretical maximum latency and memory overhead of the model combinations used in main experiments, as shown in the Table 4. The values 'Ratio' represent the additional overhead introduced by SLM relative to LLM (e.g., for combination Qwen2-0.5B & 7B, the KV cache of SLM is 1/4.67 that of the LLM).

Table 4: Quantitative analysis of SLM relative to LLM in latency and KV cache usage

| Ratio between SLM and LLM | Qwen2-0.5B & Qwen2-7B | Qwen2.5-0.5B & Qwen2.5-14B | Qwen2-7B & Qwen2-72B | LLAMA 3.2-1B & LLAMA 3.1-8B |
|---|---|---|---|---|
| Latency | 1/3.57 | 1/6.52 | 1/4.47 | 1/3.19 |
| KV cache | 1/4.67 | 1/16.0 | 1/5.71 | 1/4.0 |

We propose three deployment-level optimizations to effectively reduce the computational and memory overhead of the SLM, enhancing the practicality of SmallKV: (1) integration with speculative decoding, (2) compression of the KV cache for the SLM, and (3) layer skipping (i.e., early stopping) during the attention mapping process.

(1) The integration of speculative decoding can eliminate the forward pass of the SLM, significantly reduce latency introduced by the SLM. Current speculative decoding algorithms yield 3-5x speedups, which would further enhance performance.

(2) The high attention sparsity inherent in SLM enables near-lossless reduction in memory usage through SLM KV cache compression. As shown in the Table 5, we fix the KV cache budget of the LLM (Qwen2-7B) at 20% and increase the compression ratio of the SLM (Qwen2-0.5B) to evaluate the LLM's accuracy on BBH (SmallKV ACC = 0.436 at 20% KV cache budget). Under the same memory constraint (for SmallKV we count both SLM and LLM cache), the performance of SmallKV (ACC = 0.462 at 40% KV cache budget for SLM and 20% for LLM) outperforms H2O (ACC = 0.38 at 30% KV cache budget). Note that while SLM compression may slightly impair saliency shift compensation, the trade-off remains favorable in terms of overall efficiency.

(3) In an independently deployed scenario, the SLM is not required to generate draft tokens, enabling the adoption of an early layer stopping strategy. This approach significantly reduces both latency and memory overhead. We fix the LLM's KV cache budget at 20% and progressively reduce the number of layers utilized in the SLM for attention mapping. The results shown in Table 6 indicate that stopping at layer 20 (83% of total layers) incurs no loss in accuracy (0.436). Furthermore, even when halting at layer 16 (67%), the accuracy of SmallKV remains superior to that of H2O.

Table 5: Accuracy of LLM under different KV cache budgets of SLM

| KV Cache Budget of SLM | 20% | 40% | 60% | 80% | Full |
|---|---|---|---|---|---|
| Accuracy (BBH) | 0.2873 | 0.4416 | 0.4624 | 0.4642 | 0.4359 |

Table 6: Accuracy of LLM under different stop layers of SLM attention mapping

| Stop Layer of SLM | 24 (Full) | 20 | 18 | 16 | 12 | H3O |
|---|---|---|---|---|---|---|
| Accuracy (BBH) | 0.436 | 0.436 | 0.390 | 0.359 | 0.335 | 0.318 |

Based on the implementation optimization of (2) and (3), the SLM's actual KV cache usage is calculated as: (1 / 4.67) × 40% (SLM compression) × (20 / 24) (early stopping) ≈ 7.14% of the full LLM KV cache.

# B    Extended Discussion of Attention Similarity between Different-scale LLMs

The attention similarities between different-scale LLMs within same series are from our empirical observations. We first present further quantified results regarding attention similarity. Specifically, on the first 200 contexts from the Wikitext-v2 dataset, we record the cosine similarity between each attention head of the LLM (Qwen2-7B) and its corresponding attention matrices in SLM (Qwen2-0.5B), which is then illustrated as a heatmap in Figure 8. As can be seen from the figure, the majority of the attention heads of LLM find very similar counterparts in the SLM across different contexts. Statistical analysis of the heatmap shows that 78.22% of the elements have values exceeding 0.9, 96.21% of the elements exceed 0.8, and only 0.02% have similarities lower than 0.5.

We speculate that there are many factors that cause this similarity between different-scale LLMs within same series: 1) Although they differences in configuration parameters such as the number of layers and hidden dimensions, they share the same model architecture. 2) They are pre-trained on the similar corpus [42]. 3) Some smaller models are distilled from larger models [36]. The aforementioned factors contribute to different-scale LLMs learning similar semantic space representations at different level in the shallow decode layers. For deeper layers, previous researches [43, 13] have indicated that LLMs exhibit more attention redundancy, meaning these layers tend to have simpler attention patterns that are easy to match for the SLM, which can be evidenced by the lighter color in latter part of Figure 8.

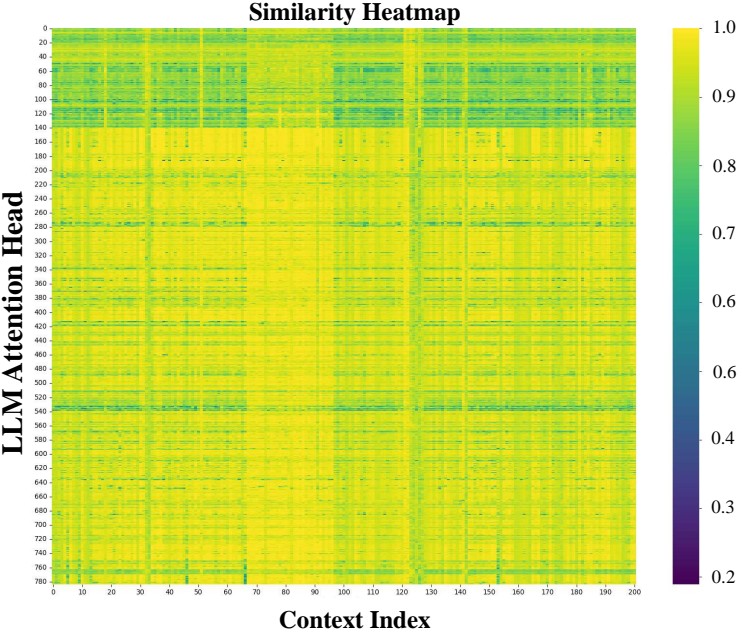

Figure 8: Similarity heatmap between Qwen2-0.5B and Qwen2-7B on 200 Contexts from Wikitext-v2. The x-axis represents different contexts, and the y-axis represents the attention heads of the LLM (Qwen2-7B). Each cell's color intensity indicates the cosine similarity of accumulated attention scores between the corresponding attention matrices of the SLM (Qwen2-0.5B) and the LLM.

To analyze the attention similarity between different SLMs and LLMs, we first visualize the attention similarity among Qwen2-7B, Qwen2-0.5B, and Llama3-1B on the Wikitext-2 dataset (as shown in Figure 9), where the y-axis represents the cumulative attention score per token across all attention heads. Results show that, due to similar training corpora and linguistic priors, models exhibit certain commonalities in their attention patterns. However, the cosine similarity between Qwen2-7B and Qwen2-0.5B reaches 0.9622, while that between Qwen2-7B and Llama3-1B is only 0.6917. This indicates that attention similarity is significantly higher within the same series models.

We further conduct a comprehensive analysis of attention pattern similarity between SLMs and LLMs across diverse models, scales, benchmarks, and multi-turn dialogue scenarios. Due to attention sinks,

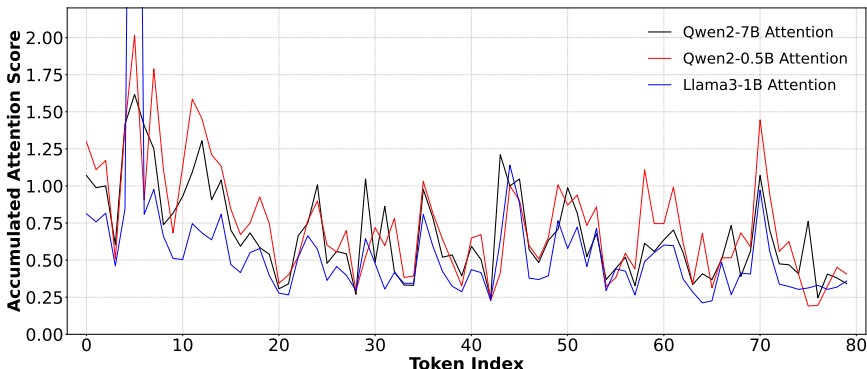

Figure 9: Accumulated attention scores of Qwen2-7B, Qwen2-0.5B, and Llama3-1B (Selecting a random segment from the middle of the tokens to avoid attention sink).

we removed the first few tokens when computing similarities to avoid too high similarity (e.g., 0.99), ensuring meaningful and comparable results. The results are represented in Table 7 and Table 8.

Table 7: Similarity between SLM and LLM across different settings (Part 1)

| SLM | Qwen2-0.5B | Qwen2-0.5B | Qwen2-0.5B | Qwen3-0.6B | Qwen2.5-0.5B | Qwen2.5-1.5B |
|---|---|---|---|---|---|---|
| LLM | Qwen2-7B | Qwen2-7B | Qwen2-7B | Qwen3-8B | Qwen2.5-14B | Qwen2.5-14B |
| Benchmark | BBH | GSM8K | MMLU | BBH | BBH | BBH |
| Similarity | 0.85 | 0.85 | 0.98 | 0.83 | 0.83 | 0.83 |

Table 8: Similarity between SLM and LLM across different settings (Continued)

| SLM | Qwen2.5-3B | Qwen2-7B | Qwen2.5-7B | LlaMA3-2.1B | LLaMA3.2-1B |
|---|---|---|---|---|---|
| LLM | Qwen2.5-14B | Qwen2-72B | Qwen2.5-14B | Qwen2-7B | LLaMA3.1-8B |
| Benchmark | BBH | BBH | BBH | BBH | BBH |
| Similarity | 0.85 | 0.81 | 0.86 | 0.56 | 0.98 |

We evaluate the changes in similarity during multi-turn dialogue scenarios, and the results are shown in Table 9. These findings demonstrate the scalability of attention similarities to a considerable extent.

## C  Extended Analysis of Marginal Information Compensation

The effectiveness of the marginal information compensation mechanism primarily originates from two aspects: high cosine similarity constraint and low attention scores constraint. The high cosine similarity constraint derived from our observations on attention similarity detailed in Appendix B, ensuring that attention patterns between the LLM and SLM are closely aligned. Meanwhile, the low attention scores constraint focuses on replacing only those tokens with relatively lower attention scores compared to critical tokens, based on the rationale that these lower-scoring tokens contribute less significantly to representation. Here we give a derivation of the low errors of attention outputs under these two constraints:

Given an original attention vector $a \in \mathbb{R}^n$, representing a row from the attention matrix, we replace it with $\tilde{a} \in \mathbb{R}^n$. The high cosine similarity constraint is defined by ensuring that the cosine similarity between $a$ and $\tilde{a}$ is very close:

$$\cos(\theta) = \frac{a^\top \tilde{a}}{\|a\|\|\tilde{a}\|} \geq 1 - \varepsilon, \quad \text{where } \varepsilon \ll 1.$$

Table 9: Attention similarity on multi-turn dialogues.

| global | turn1 | turn2 | turn3 | turn4 | turn5 |
|--------|-------|-------|-------|-------|-------|
| SmallKV | 0.822 | 0.812 | 0.799 | 0.796 | 0.782 |

Additionally, low attention scores constraint modify elements of $a$ that are small, denoted as subset $S \subset \{1, 2, \ldots, n\}$. The perturbation can be modeled as:

$$\tilde{a} = a + e,$$

where $e$ is the perturbation vector. Given the high cosine similarity constraint, $e$ mainly aligns along or close to $a$'s direction:

$$e_i = \begin{cases} \alpha a_i + \delta_i, & i \in S \\ 0, & i \notin S \end{cases}$$

Here, $\alpha \in \mathbb{R}$ represents the scaling factor along $a$, and $\delta_i$ is a small perturbation orthogonal to $a_i$ for $i \in S$. The change in output due to the perturbation is:

$$\tilde{o} - o = (\tilde{a} - a)V = eV$$

Therefore,

$$\|\tilde{o} - o\|^2 = \|eV\|^2$$

Substituting $e_i = \alpha a_i + \delta_i$ for $i \in S$ yields:

$$\|\tilde{o} - o\|^2 = \left\| \sum_{i \in S} (\alpha a_i + \delta_i) v_i \right\|^2 \leq \sum_{i \in S} (\alpha a_i + \delta_i)^2 \|v_i\|^2 + 2 \sum_{i < j \in S} |(\alpha a_i + \delta_i)(\alpha a_j + \delta_j) v_i^\top v_j|$$

If we assume $\|v_i\|$ is bounded, then:

$$\|\tilde{o} - o\|^2 \lesssim \sum_{i \in S} (\alpha a_i + \delta_i)^2$$

Since we only modify elements $a_i$ where $i \in S$, if assume that the perturbation is proportional to $a_i$. we have:

$$\mathbb{E}[e_i^2] = \sigma^2 a_i^2 \quad \text{for } i \in S$$

Thus, the total expected squared error is:

$$\mathbb{E}\left[\|eV\|^2\right] \lesssim \sum_{i \in S} \sigma^2 a_i^2$$

Notice that:

$$\sum_{i=1}^{n} a_i = 1, \quad \text{but} \quad \sum_{i \in S} a_i^2 \ll \sum_{i=1}^{n} a_i^2$$

Because $a_i$ is small when $i \in S$, $a_i^2$ is even smaller, so:

$$\sum_{i \in S} a_i^2 \ll 1 \Rightarrow \mathbb{E}\left[\|eV\|^2\right] \ll \sigma^2$$

The impact on the final output is limited by:

$$\mathbb{E}[\|\tilde{o} - o\|^2] \lesssim \sum_{i \in S} \sigma^2 a_i^2 \ll \sigma^2$$

Therefore, under the high cosine similarity constraint, replacing low attention scores has a minimal impact on the model's output.

# D More Implementation Details

We show a simplified pseudocode below to demonstrate the implementation of SmallKV. The function `SmallKV_forward()` consists of both SLM forward process and LLM forward process. In prefill stage, they can be parallelized to achieve speedup because there is no coupling. After prefilling and obtaining attention matrices in each model, the function `establish_sim_match()` is used to calculate similarity and record matching relations for subsequent steps. During the decoding, the SLM first performs its forward and updates its attention matrices. Subsequently, the LLM executes forward pass. In parallel with this process, the KV cache of the LLM is updated based on the updated attention matrices from the SLM, which is handled by the function `update_kv()`. In the forward of LLM, the standard attention forward is replaced by the function `SmallKV_attention_forward()`. Within this function, two key operations are performed: saliency shift compensation and marginal information compensation.

Listing 1: SmallKV implementation pseudocode

```
def SmallKV_forward(...):
  if prefill:
    # check the token lenth for similarity matching
    if token_lenth_meet_requirement():
      # parallel prefill for slm and llm
      slm_logits, slm_attn, slm_kv, ... =
      threading(target=forward, args=(slm, return_attn=True))
      llm_logits, llm_attn, llm_kv, ... =
      threading(target=forward, args=(llm, return_attn=True))
      # establish similarity matching
      sim_match = establish_sim_match(slm_attn, llm_attn)
      # replace attention of the llm to SmallKV attention
      llm.attention_forward = SmallKV_attention_forward
      return llm_logits
  else:
      ..., slm_attn, ... = forward(slm, return_attn = True)
      # Parallelly update KV cache
      threading(target=update_kv, args=(llm_kv, slm_attn, ...))
      llm_logits, llm_kv ...forward(llm, return_attn = False)
      return llm_logits

def establish_sim_match(slm_attn, llm_attn, ...):
  sim = calculate_pairwise_similarity(slm_attn, llm_attn)
  return sim.max(return_index)

def update_kv(llm_kv, sim_match, slm_attn, kv_cache_budget, ...):
  com_attn_slm = select(slm_attn, sim_match, layer_idx)
  selected_KV_cache = select(llm_kv, index=com_attn_slm.sum.topk)
  marg_V_cache = select(llm_kv, index=com_attn_slm.sum.top(n-k)
  offload_CPU(llm_kv,selected_K_cache,selected_V_cache,sub_V_cache)
  load_GPU(llm_kv,selected_K_cache,selected_V_cache,sub_V_cache)

def SmallKV_attention_forward(layer_idx, ...):
  saliency_compensation_attn =
  flash_attention(query, selected_K_cache, selected_V_cache, ...)
  com_attn_slm = select(slm_attn, sim_match, layer_idx)
  marginal_compensation_attn = matmul(com_attn_slm, sub_V_cache)
  return saliency_compensation_attn + marginal_compensation_attn
```

# E Limitation

There are also some limitations in our approach. First, the similarity of attention patterns between the LLM and SLM cannot be guaranteed with absolute precision, which means that our method is not lossless on LLM performance. Besides, although we have analyzed potential reasons of attention similarity, we are currently unable to provide a formal theoretical proof to rigorously support these observations. Lastly, while the assisted SLM introduced by SmallKV can share overhead with speculative decoding, this additional cost still cannot be ignored when SmallKV works alone.

# F    Case Study

Given the same prompt text in a BBT Test, we visualize the output generated by Qwen2-7B under different KV cache compression methods, including the baseline of full cache, $H_2O$ [51], Pyramid-Infer [43], and our SmallKV. The results are shown in Figure 10. With only 10% of the KV cache retained, SmallKV produces outputs that are highly similar to those generated with the full cache, both follow the "Let's think step by step." instruction and calculates according to the steps provided in the prompt examples. In contrast, $H_2O$ and PyramidInfer lose their ability to follow instructions and learn from the given reasoning patterns due to compression. Even worse, they invente new words not present in the original task description during the alphabetical ordering task (e.g., "aspect"). Moreover, $H_2O$ produces meaningless repeated content at the end of its answer.

**Model Input**

Please reference the following examples to answer the question.
Q: Sort the following words alphabetically: List: oven costume counterpart
A: Let's think step by step.
The first letter: "oven": "o" (15). "costume": "c" (3). "counterpart": "c" (3). We now have: (3) ["costume" ? "counterpart"] < (15) "oven". Now let's sort this subpart ["costume" ? "counterpart"] by looking at their second letters.
The second letter: "costume": "o" (15). "counterpart": "o" (15). We now have: (15) ["costume" ? "counterpart"]. Now let's sort this subpart ["costume" ? "counterpart"] by looking at their third letters.
The third letter: "costume": "s" (19). "counterpart": "u" (21). We now have: (19) "costume" < (21) "counterpart". Hence, we have ["costume" < "counterpart"] < "oven". So the answer is costume counterpart oven.
……

Question: Sort the following words alphabetically: List: thrill splutter panicking scorch same dot prod obstetric malton onus drumhead delmarva barn embezzle it&t damp guru subsist entirety greene

**Qwen2-7B Full Cache Output**

Let's think step by step.
The first letter: "thrill": "t" (20). "splutter": "s" (19). "panicking": "p" (16). "scorch": "s" (19). "same": "s" (19). "dot": "d" (4). "prod": "p" (16). "obstetric": "o" (15). "malton": "m" (13). "onus": "o" (15). "drumhead": "d" (4). "delmarva": "d" (4). "barn": "b" (2). "embezzle": "e" (5). "it&t": "i" (9). "damp": "d" (4). "guru": "g" (7). "subsist": "s" (19). "entirety": "e" (5). "greene": "g" (7). We now have: (2) ["barn"? "dot"] < (4) ["delmarva"? "drumhead"] < (5) ["embezzle"? "it&t"] < (7) ["guru"? "greene"] < (9) "it&t" < (11) ["obstetric"? "onus"

**Qwen2-7B H₂O 10% Cache Output**

1. "aspect": "a" (1). "barn": "b" (2). "census": "c" (3). "costume": "c" (3). "credulity": "c" (3). "drumhead": "d" (4). "embezzle": "e" (5). "entirety": "e" (5). "greene": "g" (7). "hypochlorite": "h" (11). "mugho": "m" (13). "newt": "n" (14). "obstetric": "o" (15). "parthia": "p" (16). "prod": "p" (16). "scorch": "s" (19). "same": "s" (19). "seismography": "s" (19). "splutter": "s" (19). "subsist": "s" (19). "thrum": "t" (20). "tremble": "t" (20). "tremble": "t" (20). "tremble": "t" (20). "tremble": "t" (20). "tremble": "t" (20). "tremble": "t" (

**Qwen2-7B PyramidInfer 10% Cache Output**

1. "aspect" 2. "arson" 3. "census" 4. "drumhead" 5. "embezzle" 6. "entirety" 7. "guru" 8. "it&t" 10. "malton" 11. "mugho" 12. "panicking" 13. "prod" 14. "scorch" 15. "same" 16. "seismography" 17. "splutter" 18. "subsist" 19. "thril" 20. "tarnish" 21. "universe" 22. "verity" 23. "vulgar" 24. "wagtail" 25. "wagtail"
So the answer is arson aspect census drumhead embezzle entirety guru it&t malton mugho panicking prod scorch same thril tarnish universe verity vulgar wagtail wagtail.

**Qwen2-7B SmallKV 10% Cache Output**

Let's think step by step.
The first letter: "thrill": "t" (20). "splutter": "s" (19). "panicking": "p" (16). "scorch": "s" (19). "same": "s" (19). "dot": "d" (4). "prod": "p" (16). "obstetric": "o" (15). "malton": "m" (13). "onus": "o" (15). "drumhead": "d" (4). "delmarva": "d" (4). "embezzle": "e" (55). "it&t": "i" (9). "damp": "d" (4). "guru": "g" (7). "subsist": "s" (19). "entirety": "e" (55). "greene": "g" (7). We now have: (4, 4, 7, 7, 9, 13, 13, 19, 19, 20. The words sorted alphabetically are: "damp", "dot", "entirety", "embezzle", "guru", "it&t", "obstetric", "panicking", "prod", "scorch

Figure 10: Visualized outputs of one generation example with Qwen2-7B. Results are compared between the baseline model with full cache, H₂O [51], PyramidInfer [43], and our SmallKV.

