# OpenReview forum: "SmallKV: Small Model Assisted Compensation of KV Cache Compression for Efficient LLM Inference"
_NeurIPS.cc/2025/Conference — NeurIPS 2025 spotlight_

### Official Review · Reviewer_Dh4K · 2025-06-27

**Clarity:** 3
**Significance:** 3
**Originality:** 3
**Rating:** 4
**Confidence:** 4

**Summary:**

In this paper, authors propose a new KV cache compression method named SmallKV, based on the high similarity of attention matrices between LLMs with different scales. SmallKV uses small models to assist the large model in perceiving globally important information
 of attention and approximates the attention score for large models. SmallKV could solve the saliency shift problem and the marginal information over-compression problem. Extensive experiments on benchmarks demonstrate the effectiveness of SmallKV.

**Questions:**

1. The implementation is not detailed. Given that using 8 A100 GPUs, which parallelism did you use?
2. What's the exactly memory usage for your experiments?
3. Given that your complexity is still O(n^2) for small models, It would have a theorical compression upper bound for your method, can you show this?
4. In Figure 1(c), can you add the comparison to LLaMA3.2-1B? It would much enhance your conclusion.
5. Why does BHH accuracy much worser for Qwen2-7B with Qwen2-72B in Figure 5？
6. Can this method be implemented together with Speculative Sampling such as Eagle?
7. How does the magnitude of attention pattern divergence between SLMs and LLMs correlate with downstream performance?

If my concerns are solved, I would raise my score.

**Ethical Concerns:**

["NO or VERY MINOR ethics concerns only"]

**Final Justification:**

My main concern about this paper has been solved.
- A substantial drop in attention similarity between models from different families.
- The ablation study of memory cost is sufficient.

So I raise my rating to Borderline accept.

**Limitations:**

- It's hard to be implemented on edge-device.
- The memory usage increased is not small.
- It have a benefit upper bound compared to other evict-based method.
- The system optimization for such method is hard.

**Quality:**

3

**Strengths And Weaknesses:**

Strengths:
- This paper is well-written and easy to follow.
- The insight and observation is new and interesting for me.
- The theory analysis is rich and make sense.
- It can be easily implemented with other methods.

Weakness:
- There's no memory cost analysis experiments, I'm skeptical about the memory cost.
- The requirement for SmallKV to use smaller models in the same series may limit its applicability. It remains unclear whether this approach can effectively enable efficient KV caching for SLMs.
- The experiments to approve highly consistent attention patterns between LLMs with different scaling only contain the same serial models, missing negative comparison.

---

> ### Author Rebuttal · Authors · 2025-07-30
>
> Thank you for your detailed review and valuable feedback! Below, we address the comments you raised:
>
> **Q1: The implementation is not detailed. Given that using 8 A100 GPUs, which parallelism did you use?**
>
> **R**: We utilize pipeline parallelism by setting the device_map = "auto" feature from the Hugging Face Accelerate library. The configuration will be explicitly described in the revised manuscript.
>
> **Q2: What's the exactly memory usage for your experiments?**
>
> **R**: We acknowledge that SmallKV introduces additional memory overhead, primarily from the KV cache of the small model. Specifically, for the four model combinations used in our main experiments, the theoretical maximum memory consumption ratio between the SLM's and LLM's KV cache is shown in the table below. The values 'Ratio' represent the additional overhead introduced by SLM relative to LLM (e.g., for combination Qwen2-0.5B & 7B, the KV cache of SLM is 1/4.67 that of the LLM).
>
> |Ratio between SLM and LLM|Qwen2-0.5B & Qwen2-7B|Qwen2.5-0.5B & Qwen2.5-14B|Qwen2-7B & Qwen2-72B|LLAMA 3.2-1B & LLAMA 3.1-8B|
> |-|-|-|-|-|
> |KV cache|1/4.67|1/16.0|1/5.71|1/4.0|
>
> However, we emphasize that SmallKV remains highly practical, and the overhead of the SLM can be significantly reduced through deployment-level optimizations.We propose three key optimizations: (1) integration with speculative decoding (SP), (2) KV cache compression for the SLM, and (3) layer skipping (i.e., early stopping) during attention mapping.
>
> The high attention sparsity observed in the SLM enables nearly lossless reductions in memory usage via KV cache compression (reduce to 40% of SLM cache). As shown in the table below, we fix the KV cache budget of the LLM (Qwen2-7B) at 20% and increase the compression ratio of the SLM (Qwen2-0.5B) to evaluate the LLM’s accuracy on BBH (SmallKV ACC = 0.436 at 20%). Under the same memory constraint (for SmallKV we count both SLM and LLM cache), the performance of SmallKV (ACC = 0.462 at 40% for SLM and 20% for LLM) outperforms H2O (ACC = 0.38 at 30%). Note that while SLM compression may slightly impair saliency shift compensation, the trade-off remains favorable in terms of overall efficiency.
>
> |KV Cache Budget of SLM|20%|40%|60%|80%|Full|
> |-|-|-|-|-|-|
> |Accuracy (BBH)|0.2873|0.4416|0.4624|0.4642|0.4359|
>
> Furthermore, in the independently deployed scenario, the SLM does not need to generate draft tokens. This allows SLM to adopt an early layer stopping strategy, significantly reducing its latency and memory overhead. We fix the LLM’s KV cache budget at 20% and gradually reduce the number of layers used in the SLM for attention mapping. The results indicate that stopping at layer 20 (83% of total layers) incurs no loss in accuracy (0.436). Furthermore, even when halting at layer 16 (67%), the accuracy of SmallKV remains superior to that of H2O.
>
> |Stop Layer of SLM |24 (Full)|20|18|16|12|H2O|
> |-|-|-|-|-|-|-|
> | Accuracy (BBH)|0.436|0.436|0.390|0.359|0.335|0.318|
>
> Finally, we would like to highlight that, to date, optimizations for LLM inference have primarily focused on two directions: (1) speculative decoding for GPU utilization optimization and (2) KV cache compression for memory and time efficiency. This work represents the first attempt to efficiently combine both approaches. Moving forward, we can further leverage advancements in these two areas to enhance the performance of a unified system.
>
> **Q3：Given that your complexity is still O(n^2) for small models, It would have a theorical compression upper bound for your method, can you show this?**
>
> **R**: We apologize for not fully grasping the intent of your question. Our interpretation is that, this refers to the overall compression upper bound caused by the need to save SLM full cache. Based on this, we present the theoretical maximum memory consumption of the small model in the SLM-to-LLM KV cache ratio table in Q2, and further propose two optimization techniques to reduce this overhead. The experimental results reflect a practical upper bound on compression. For example, for Qwen2-0.5B & 7B combination, when the LLM's KV cache budget is 20%, the SLM’s actual KV cache usage—under accuracy-preserving conditions—is calculated as:(1/4.67)×40%(SLM compression)×(20/24)(early stopping)≈ 7.14% of the full LLM KV cache.
>
> If this answer does not fully address your question, we welcome your feedback and clarification. We are willing to provide you with more detailed explanation as needed.
>
> **Q4：In Figure 1(c), can you add the comparison to LLaMA3.2-1B? It would much enhance your conclusion.**
>
> **R**: Thanks for your valuable suggestions. We have successfully added the comparison to LLaMA3.2-1B. Due to restrictions on sharing images or links here, we instead quantify this observation using average cosine similarity on the Wikitext-2 dataset. Notably, we observe a substantial drop in attention similarity between models from different families. The average similarity between Qwen2-0.5B and Qwen2-7B is 0.947, while the similarity between LLaMA3.2-1B (used as the SLM) and Qwen2-7B (the LLM) drops to 0.632. (To account for tokenizer differences, we compute pairwise longest common subsequences to align tokens and calculate cosine similarity only over matched tokens.)
>
> **Q5：Why does BHH accuracy much worser for Qwen2-7B with Qwen2-72B in Figure 5？**
>
> **R**: To investigate this issue, we measure the average attention similarity between large and small models across several model combinations on the BBH dataset, as shown in the table below. The results show a positive (nonlinear) correlation between attention similarity and task accuracy. So the lower accuracy is because of the lower similarity of Qwen2-7B and Qwen2-72B. Please refer to Q7, the discussion about attention similarity and SLM scale.
>
> |SLM|Qwen2-0.5B|Qwen2.5-0.5B|Qwen2-7B|LLaMA3.2-1B|
> |-|-|-|-|-|
> |LLM|Qwen2-7B|Qwen2.5-14B|Qwen2-72B|LLaMA3.1-8B|
> |Similarity|0.85|0.83|0.81|0.98|
>
> **Q6: Can this method be implemented together with Speculative Sampling such as Eagle?**
>
> **R**: The drafting method in Speculative Sampling can be classified into Independent Drafting and Self-Drafting[1], SmallKV is compatible with Independent Drafting, where a smaller model from the same series is uesd as the drafter [2-3]. In contrast, Eagle follows the Self-Drafting paradigm, utilizing the final hidden states of a large model as input features to train an external transformer layer for drafting. So Eagle is not compatible with SmallKV.
>
> [1] Unlocking Efficiency in Large Language Model Inference: A Comprehensive Survey of Speculative Decoding
>
> [2] Accelerating large language model decoding with speculative sampling
>
> [3] Cascade speculative drafting for even faster llm inference.
>
> **Q7: How does the magnitude of attention pattern divergence between SLMs and LLMs correlate with downstream performance?**
>
> **R**: As demonstrated in Q5, low attention similarity leads to a relative performance drop. We also test the attention similarity and the performance (under 10% KV Cache Budget) across four SLMs of varying scales, using Qwen2.5-14B as LLM. The results, summarized in the table below, reveal a clear trend: as the size of the SLM increases, both attention similarity and task performance consistently improve.
>
> |SLM|Qwen2.5-0.5B|Qwen2.5-1.5B|Qwen2.5-3B|Qwen2.5-7B|
> |-|-|-|-|-|
> |Similarity|0.832|0.834|0.846|0.855|
> |Accuracy (BBH)|0.402|0.457|0.483|0.486|
>
> Therefore, a reasonable conclusion is that attention similarity is positively correlated with downstream task performance. Ideally, a similarity score of 1.0 would yield optimal predictive KV cache compression. However, it is important to recognize that this relationship is influenced by multiple factors, including model scale, the evaluation tasks, and the specific dynamics of KV cache compression. As a result, while the correlation is evident, it is difficult to precisely quantify the extent to which divergence in attention patterns affects downstream performance.
>
> We would also like to offer some clarification regarding the discussed weaknesses:
>
> **Weaknesse1: There's no memory cost analysis experiments, I'm skeptical about the memory cost.**
>
> **R**: We provided a detailed quantitative analysis of memory cost in Q2, and we have also implemented two methods to reduce memory overhead: KV cache compression for the small model and early stopping (see Q2 for details). We hope these explanations and solutions adequately address your concerns.
>
> **Weaknesse2: The requirement for SmallKV to use smaller models in the same series may limit its applicability. It remains unclear whether this approach can effectively enable efficient KV caching for SLMs.**
>
> **R**: Developing multiple models of varying sizes to accommodate the diverse range of central and edge devices has become a prevailing trend in mainstream LLM development (e.g., the Llama, Qwen, and Phi-3 families). Furthermore, even when a directly suitable small model is not readily available, it can be effectively obtained through distillation by training a compact model from a larger one.
>
> SmallKV has been proven to effectively enable efficient KV caching for SLMs. As response in Q2, we have implemented KV cache compression for SLM, specifically aiming to reduce the associated memory overhead.
>
> **Weaknesse3:
> The experiments to approve highly consistent attention patterns between LLMs with different scaling only contain the same serial models, missing negative comparison.**
>
> **R**: In Q4, we conduct a negative comparison by measuring attention similarity between models from different families. The results show that the attention similarity between models from different series drops significantly. This suggests that, models from different series may not be suitable for use with SmallKV in achieving effective accuracy compensation.
>
> We hope this response has adequately addressed your concerns. We look forward to the opportunity for additional discussion during the review process.

---

> > ### Comment · Reviewer_Dh4K · 2025-08-03
> >
> > Thank you for the rebuttal. The ablation study about the memory usage and negative comparison is sufficient. So I will raise my rating. Please include them in your final version paper.
> >
> > By the way, I wonder if the draft models in Eagle can show a similar phenomenon to the SLM? If you train a draft model by distillation, can this help to merge your method with Eagle?

---

> > > ### Author Response · Authors · 2025-08-03
> > >
> > > We sincerely thank you for your constructive comments and feedback, which have greatly assisted us in enhancing the completeness of our paper. We will incorporate the supplementary experiments presented in this rebuttal into the final version of our manuscript.
> > >
> > > Regarding the question you raised, combining SmallKV with Eagle to achieve an efficient system is an interesting direction. The primary issue we have identified is not merely that Eagle requires additional training, but rather that it adds only one autoregressive head for drafting, which provides limited information of attention patterns. Rich source attention matrices are fundamental to the effectiveness of SmallKV. For instance, we observe that early stopping in shallow layers (i.e. 12/24) of small models leads to performance degradation in SmallKV, which, at its core, also reduces the richness of the source attention matrices. However, above points remain theoretical. We will consider your insightful suggestion in our future work integrating speculative decoding, where we plan to explore ways to improve the performance when combining SmallKV with Eagle.

---

### Official Review · Reviewer_rx3H · 2025-06-28

**Clarity:** 3
**Significance:** 2
**Originality:** 2
**Rating:** 4
**Confidence:** 3

**Summary:**

The paper presents a design named SmallKV, a method for compressing Key-Value (KV) caches in large language models (LLMs) to enable efficient inference in long-context scenarios. SmallKV leverages a small language model (SLM) to address two critical limitations of existing token-level eviction methods: (1) the saliency shift issue, where dynamic changes in token importance during decoding lead to suboptimal permanent eviction, and (2) the marginal information over-compression issue, where marginally important tokens, collectively significant to performance, are treated the same as unimportant ones. SmallKV introduces saliency shift compensation, utilizing the SLM’s full cache to dynamically adjust LLM eviction strategies, and marginal information compensation, which approximates the marginal tokens’ Key cache using SLM attention scores while retaining their Value cache.

**Questions:**

Besides the weaknesses above:

The main experimental results lack error bars or confidence intervals. Please provide a statistical analysis and specify the method used for computation.

Figure 7 illustrates that larger SLMs yield limited performance gains, accompanied by increased computational cost. Please analyze the trade-off between SLM scale and performance, specifying optimal SLM size criteria.

The selection of thresholds for marginal tokens (K and P-K) is not well justified. Please provide a theoretical or empirical analysis of threshold selection and its impact on performance.

**Ethical Concerns:**

["NO or VERY MINOR ethics concerns only"]

**Final Justification:**

The authors' response has addressed most of the points (regarding the method's foundational assumption, performance overhead, and the justification for confident design choices). The new empirical data demonstrating high SLM-LLM attention similarity across diverse models, tasks, and even multi-turn dialogues provides strong support for the method's core assumption, mitigating my concerns about its generalizability. I improved my assessment of the paper's contribution.

**Limitations:**

yes

**Quality:**

2

**Strengths And Weaknesses:**

Strengths:

SmallKV utilizes an SLM to enhance KV cache compression, incorporating saliency shift and marginal information compensation mechanisms to address the limitations of traditional eviction methods. The insight of high attention similarity between SLM and LLM provides a novel perspective.

Jaccard similarity matching of SLM-LLM attention matrices, dynamic eviction, and marginal token approximation enable efficient compression.

Weaknesses:

The method's core assumption—that attention patterns between different-scale LLMs are highly similar—is an empirical observation. The authors acknowledge they cannot provide a formal theoretical proof to guarantee this similarity, which raises questions about the method's generalizability across all models and tasks. Therefore, the technique is not lossless, and its performance depends on the quality of the attention matching.

The results presented in Table 2 show that even when using a 0.5B model, your method does not hold an advantage over H2O in terms of TPOT. This suggests that the TPOT overhead is a potential concern, especially if the SLM's parameter size were to increase.
The chosen baseline, H2O (2023), is somewhat dated. It would be more convincing if the authors could demonstrate that their method has advantages over more recent SOTA works.

The paper's approach to using SLM to capture global attention views may struggle to ensure comprehension of complex sentences, as the evaluation is limited to simple mathematical tasks, such as GSM8K. However, current research trends focus on enabling LLMs to tackle more challenging problems, such as those in AMIE  and AMC.

---

> ### Author Rebuttal · Authors · 2025-07-30
>
> Thank you for your detailed review and valuable feedback! Below, we address the comments you raised:
>
> **Q1: The main experimental results lack error bars or confidence intervals. Please provide a statistical analysis and specify the method used for computation.**
>
> **R**: All baseline methods, including ours, are conducted in a deterministic setting, which is the standard evaluation approach adopted in prior KV cache research. For our experiments, we set the model's temperature to 0, leading to greedy decoding, where the model always selects the token with the highest probability at each step. This results in fully deterministic outputs for any given input and model configuration.
>
> When evaluating on standardized benchmarks with fixed prompts and model parameters (temperature=0), repeated runs produce identical outputs and accuracies. In such a deterministic setting, error bars or confidence intervals are not applicable, as they are designed to capture uncertainty arising from stochastic variability in results.
>
> **Q2: Figure 7 illustrates that larger SLMs yield limited performance gains, accompanied by increased computational cost. Please analyze the trade-off between SLM scale and performance, specifying optimal SLM size criteria.**
>
> **R**: Based on the experimental results shown in Figure 7, we conclude that when the KV cache budget is relatively large, scaling the small model has a relatively minor effect on performance (less than 10%). However, SLM scaling significantly influences performance under extremely low KV cache budgets (e.g., 5%). Therefore, in practical production systems, when resources are sufficient, a larger SLM scale generally leads to higher accuracy of the LLM. Under resource-constrained conditions, a trade-off between efficiency and performance must be made. For instance, in scenarios requiring extreme compression, a more suitable choice would be to increase the scale of the SLM. Conversely, when the compression ratio is small, the SLM's size can be appropriately reduced. This represents a trade-off between resources, performance, and application scenarios.
>
> Furthermore, SmallKV is compatible with speculative decoding in deployment. Therefore, the choice of SLM scale is highly deployment-dependent, and should be determined jointly based on the accuracy requirements of the draft model in speculative decoding and the available memory resources on the target hardware.
>
> **Q3: The selection of thresholds for marginal tokens (K and P-K) is not well justified. Please provide a theoretical or empirical analysis of threshold selection and its impact on performance.**
>
> **R**: Indeed, using a fixed threshold for marginal tokens may not always lead to optimal results. In our evaluation, we adopted the same experimental parameters as other baselines to ensure comparability. However, a key advantage of SmallKV lies in the SLM's ability to adaptively identify which tokens are most important.
>
> Therefore, in deployment, we can dynamically define the proportions of saliency shift compensation and marginal tokens compensation based on the recall rate of attention scores from the small model, as illustrated in Figure 2. For instance, we consider the critical tokens to be those accounting for the top 80% of cumulative attention scores, while marginal tokens are defined as those within the subsequent 15% range (i.e., from 80% to 95% of the cumulative sum). This adaptive approach allows us to fine-tune the balance between saliency and marginal tokens to better suit the specific demands of different tasks.
>
> We would also like to offer some clarification regarding the discussed weaknesses:
>
> **Weakness 1: The method's core assumption—that attention patterns between different-scale LLMs are highly similar—is an empirical observation. The authors acknowledge they cannot provide a formal theoretical proof to guarantee this similarity, which raises questions about the method's generalizability across all models and tasks. Therefore, the technique is not lossless, and its performance depends on the quality of the attention matching.**
>
> **R**: It is hard to provide a rigorous theoretical proof, instead, we have conducted extensive tests on the attention similarity between large and small models across various scenarios to address this concern. Due to attention sinks, we removed the first few tokens when computing similarities to avoid too high similarity (e.g., ~0.99), ensuring meaningful and comparable results.
>
> Firstly, we measured the similarity across different model combinations in our main experiments on the BBH task:
>
> | SLM | Qwen2-0.5B | Qwen2.5-0.5B | Qwen2-7B | LLaMA3.2-1B |
> |-------------|-------------|-------------|-------------|-------------|
> | LLM | Qwen2-7B | Qwen2.5-14B | Qwen2-72B | LLaMA3.1-8B |
> | Similarity | 0.85 | 0.83 | 0.81 | 0.98 |
>
> Additionally, we evaluated the attention similarity between Qwen2 7B and Qwen2 0.5B across multiple tasks:
>
> | Benchmark | BBH |  GSM8K |  MMLU |
> |-------------|-------------|-------------|-------------|
> | Similarity | 0.85 | 0.85 | 0.98 |
>
> Furthermore, we examined the changes in similarity during multi-turn dialogue scenarios:
>
> |       | turn1 | turn2 | turn3 | turn4 | turn5 |
> |-------|-------|-------|-------|-------|-------|
> | global| 0.822 | 0.812 | 0.799 | 0.796 | 0.792 |
> | SmallKV| 0.822 | 0.809 | 0.792 | 0.791 | 0.787 |
>
> These findings demonstrate the scalability of attention similarities to a considerable extent.
>
> **Weakness 2: The results presented in Table 2 show that even when using a 0.5B model, your method does not hold an advantage over H2O in terms of TPOT. This suggests that the TPOT overhead is a potential concern, especially if the SLM's parameter size were to increase. The chosen baseline, H2O (2023), is somewhat dated. It would be more convincing if the authors could demonstrate that their method has advantages over more recent SOTA works.**
>
> **R**: In the dynamic compensation process during decoding, SmallKV requires an additional forward pass through the SLM, which cannot be fully parallelized as in the prefill stage, which leads to a slightly lower TPOT compared with H2O. However, the partial compatibility with Flash Attention makes SmallKV achieving higher overall throughput.
>
> The decoding efficiency overhead can be effectively mitigated in deployment by integrating SmallKV with speculative decoding, where the SLM forward pass is already part of the pipeline and does not contribute to additional latency. State-of-the-art speculative decoding methods commonly achieve 3–5× speedups.
>
> We emphasize that the primary goal of SmallKV is not to propose a new KV cache compression algorithm, but rather to provide a general compensation mechanism that improves the accuracy of existing eviction-based methods. Any resulting inference acceleration is a beneficial side effect. This also addresses your second question.
>
> We compare against H2O and PyramidInfer because our current implementation builds upon their cumulative attention score-based eviction frameworks. Importantly, SmallKV is not mutually exclusive with other KV cache compression techniques. SmallKV is a accuracy compensation mechanism designed based on the principle of attention sparsity in LLMs, specifically targeting attention score-based KV cache compression methods. As such, SmallKV is complementary and can be applied to other SOTA KV compression approaches to further enhance their accuracy.
>
> **Weakness 3: The paper's approach to using SLM to capture global attention views may struggle to ensure comprehension of complex sentences, as the evaluation is limited to simple mathematical tasks, such as GSM8K. However, current research trends focus on enabling LLMs to tackle more challenging problems, such as those in AMIE and AMC.**
>
> **R**: Firstly, we have supplemented the attention pattern similarity results under a broader range of scenarios in response to weakness 1. These additional tests provide a more comprehensive evaluation of the scalability across different scenarios.
>
> Secondly, we would like to emphasize that the primary goal of KV Cache compression is to preserve the foundational capabilities of the model rather than to extend its limits. Benchmarks such as AMIE are often used to explore the upper bounds of model capabilities. Our extensive and comprehensive evaluation in paper already covers a wide array of common LLM tasks, including logical reasoning, mathematics, coding, multi-turn dialogue, and long-context understanding. And the results show that SmallKV ensures improvement in accuracy across diverse tasks. We believe that the accuracy compensation strategy of SmallKV can also be applied to more challenging tasks, helping models improve inference accuracy while reducing memory footprint.
>
> We hope this response has adequately addressed your concerns. We look forward to the opportunity for additional discussion during the review process.

---

> > ### Comment · Reviewer_rx3H · 2025-08-06
> >
> > Dear authors,
> >
> > Thank you for the detailed rebuttal. The authors' response has addressed most of the points (regarding the method's foundational assumption, performance overhead, and the justification for confident design choices). The new empirical data demonstrating high SLM-LLM attention similarity across diverse models, tasks, and even multi-turn dialogues provides strong support for the method's core assumption, mitigating my concerns about its generalizability. I improved my assessment of the paper's contribution.

---

> > > ### Author Response · Authors · 2025-08-06
> > >
> > > Dear Reviewer rx3H,
> > >
> > > Thank you for your recognition of our work. We are delighted to receive your feedback. We will incorporate the points you raised and discussed into the final version of our paper to enhance its completeness. If you have any further questions, please feel free to comment at any time, and we will promptly address your concerns.

---

> ### Author Response · Authors · 2025-08-06
>
> We sincerely appreciate your thorough and insightful review on our work.
>
> We hope that our responses provided above address your primary concerns. Your feedback is crucial for enhancing the completeness and quality of our work.
>
> If you have any further queries or require additional information, please feel free to let us know. Thank you once again for your time and valuable input.

---

### Official Review · Reviewer_9RZw · 2025-06-29

**Clarity:** 3
**Significance:** 3
**Originality:** 3
**Rating:** 5
**Confidence:** 4

**Summary:**

Efficient management of the KV-cache is critical to mitigating resource contention during LLM inference. Existing approaches often suffer from issues such as saliency shift, where token importance dynamically changes, and marginal information overcompression, where individually less important tokens are overly compressed, degrading overall model accuracy. This paper observes that a small language model can produce attention scores similar to those of a much larger LLM. Motivated by this insight, the authors propose SmallKV, a method leveraging an SLM to guide efficient compression and eviction strategies for the KV-cache. Specifically, SmallKV uses the smaller model to track global attention patterns and approximate attention for marginal tokens, significantly improving both accuracy and inference throughput over existing approaches.

**Questions:**

1. How sensitive is the effectiveness of SmallKV to the size and choice of the small model? Did you experiment with architectures significantly different from the large model?

2. Could you elaborate on why the attention patterns between the small and large models remain highly similar across scales? Are there specific conditions under which this similarity might break down?

3. How exactly were thresholds for categorizing tokens into critical, marginal, and unimportant chosen? Can you share your insight of choosing 2:1:2

**Ethical Concerns:**

["NO or VERY MINOR ethics concerns only"]

**Final Justification:**

The paper is strong, the rebuttal results alleviated my concerns, therefore I support its acceptance.

**Limitations:**

Yes

**Quality:**

3

**Strengths And Weaknesses:**

Strengths:

1. The paper clearly identifies the importance of marginal tokens and conducts the empirical experiments to validate the authors' observations.

2. The paper introduces a novel use of a smaller auxiliary model for dynamic KV-cache management.

Weaknesses:

1. The use of an auxiliary smaller model introduces additional memory and computational overhead

2. While some ablations are provided, more sensitivity experiment like varying sizes of SLM is missing

---

> ### Author Rebuttal · Authors · 2025-07-30
>
> Thank you for your detailed review and valuable feedback! Below, we address the comments you raised:
>
> **Q1: How sensitive is the effectiveness of SmallKV to the size and choice of the small model? Did you experiment with architectures significantly different from the large model?**
>
> **R**: In Section 5.5 and Figure 7, we examine the impact of small model size (ranging from 0.5B to 7B parameters) on the effectiveness of SmallKV. Our findings show that when the KV cache budget is relatively large, scaling the small model has a relatively minor effect on performance (less than 10%). However, SLM scaling significantly influences performance under extremely low KV cache budgets (e.g., 5%). However, under extremely constrained KV cache budgets (e.g., 5%), the size of the small model plays a much more significant role in performance. This suggests that an effective strategy is to select a small model of appropriate size based on the desired KV cache compression ratio.
>
> For SLMs with architectures significantly different from the large model, two main issues prevent the direct application of them in SmallKV. First, differences in tokenizers lead to mismatched dimensions and token representations in their attention matrices, making the compensation mechanism proposed in SmallKV inapplicable. Second, we measured the attention similarity between models from different series. Results show that the average cosine similarity between Llama3-1B and Qwen2-7B on the Wikitext-2 dataset is only 0.632 (note that due to vocabulary differences, we calculate similarity only on matched tokens via longest common subsequence alignment). In contrast, using Qwen2-0.5B as the SLM yields an average similarity of 0.947. This indicates that models from different series do not exhibit significant attention similarity.
>
> **Q2: Could you elaborate on why the attention patterns between the small and large models remain highly similar across scales? Are there specific conditions under which this similarity might break down?**
>
> **R**: The attention similarities between different-scale LLMs within same series are based on our empirical findings. We speculate that many factors contribute to this similarity: 1)  although models of varying scales differ in configuration parameters such as the number of layers and hidden dimensions, they share the same underlying architecture; 2) they are typically pre-trained on similar corpora; 3) in some cases, smaller models are distilled from their larger counterparts. Beyond these considerations, we believe a key contributing factor is the inherent redundancy in the attention patterns of LLMs, which enables a compact SLM to effectively approximate the attention behavior of a larger model.
>
> To explore the conditions under which attention similarity might degrade, we conducted a comprehensive evaluation across multiple datasets—including MMLU, BBH, and GSM8K—and various SLMs ranging from Qwen2.5-0.5B to Qwen2.5-7B. And the results are shown in the Table below. The results indicate that, while some variability exists, attention similarity remains consistently high across different datasets within the same model series. For contrast, we also include a negative comparison, reporting attention similarity between models from different series. Due to differences in tokenizers across series, we align tokens using pairwise longest common subsequences and compute cosine similarity only over matched tokens. Based on our current experiments, we have not observed any specific condition under which the attention similarity between models within the same series fully breaks down.
>
> |SLM|Qwen2-0.5B|Qwen2-0.5B|Qwen2-0.5B|Qwen3-0.6B|Qwen2.5-0.5B|Qwen2.5-1.5B|Qwen2.5-3B|Qwen2.5-7B|LLaMA3.2-1B|
> |-|-|-|-|-|-|-|-|-|-|
> |LLM|Qwen2-7B|Qwen2-7B|Qwen2-7B|Qwen3-8B|Qwen2.5-14B|Qwen2.5-14B|Qwen2.5-14B|Qwen2.5-14B|Qwen2-7B|
> |Benchmark|BBH|GSM8K|MMLU|BBH|BBH|BBH|BBH|BBH|BBH|
> |Similarity|0.85|0.85|0.98|0.83|0.83|0.83|0.85|0.86|0.56|
>
> **Q3: How exactly were thresholds for categorizing tokens into critical, marginal, and unimportant chosen? Can you share your insight of choosing 2:1:2**
>
> **R**: Our evaluation results are primarily based on a fixed allocation ratio, adopting the same parameter setting as H2O for fair comparison (heavy : recent  = 1:1). So in our experiment, heavy tokens : recent : compensation = 2:1: 2 (as compensate token only occupy half of cache compared to others). However, in practice, we observe that the optimal compensation ratio varies slightly depending on the task and model scale. For instance: (1) When the KV cache budget is extremely limited (e.g., 5%), allocating the majority of the budget to heavy tokens yields better performance. (2) When the scale gap between the SLM and LLM is larger (e.g., 70B vs. 7B, compared to 7B vs. 0.5B), slightly reducing the compensation ratio leads to improved accuracy.
>
> Therefore, we suggest that in practical deployment of SmallKV, the adaptive token importance estimation capability of the SLM can be leveraged to dynamically adjust the allocation ratios and thresholds. As illustrated in Section 3.2 Figure 2, the proportions of critical tokens and marginal tokens can be dynamically determined based on the recall of attention scores from the small model, tailored to different tasks. For instance, we define the critical tokens as those accounting for the top 80% of cumulative attention scores, while marginal tokens are set to the subsequent 15% (i.e., from 80% to 95% of the cumulative sum).
>
> We would also like to offer some clarification regarding the discussed weaknesses:
>
> **Weakness 1:
> The use of an auxiliary smaller model introduces additional memory and computational overhead.**
>
> **R**: We quantify the theoretical maximum latency and memory overhead of the model combinations used in main experiments, as shown in the table below. The values 'Ratio' represent the additional overhead introduced by SLM relative to LLM (e.g., for combination Qwen2-0.5B & 7B, the KV cache of SLM is 1/4.67 that of the LLM).
>
> |Ratio between SLM and LLM|Qwen2-0.5B & Qwen2-7B|Qwen2.5-0.5B & Qwen2.5-14B|Qwen2-7B & Qwen2-72B|LLAMA 3.2-1B & LLAMA 3.1-8B|
> |-|-|-|-|-|
> |Latency|1/3.57|1/6.52|1/4.47|1/3.19|
> |KV cache|1/4.67|1/16.0|1/5.71|1/4.0|
>
> We highlight the practicality of SmallKV, emphasizing how the computational overhead of the SLM can be effectively mitigated through deployment-level optimizations. To achieve this, we propose three key strategies: (1) integration with speculative decoding (SP); (2) compression of the KV cache for the SLM; and (3) layer skipping (i.e., early stopping) during the attention mapping process.
>
> Regarding latency, while the forward pass of the SLM during decoding incurs some overhead, this is mitigated by the acceleration provided through KV cache compression. As demonstrated in Section 5.3 Table 2 of the evaluation, SmallKV achieves lower latency compared to the baseline model. Additionally, the integration of SP can eliminate the forward pass of the SLM as a significant latency bottleneck. Current SP algorithms yield 3-5x speedups, further enhancing performance.
>
> Regarding memory consumption, the high attention sparsity inherent in SLM enables near-lossless reduction in memory usage through SLM KV cache compression. As shown in the table below, we fix the KV cache budget of the LLM (Qwen2-7B) at 20% and increase the compression ratio of the SLM to evaluate the LLM’s accuracy on BBH (SmallKV ACC = 0.436 at 20% budget). Under the same memory constraint (for SmallKV we count both SLM and LLM cache), the performance of SmallKV (ACC = 0.462 at 40% for SLM and 20% for LLM) outperforms H2O (ACC = 0.38 at 30%). Note that while SLM compression may slightly impair saliency shift compensation, the trade-off remains favorable in terms of overall efficiency.
>
> |KV Cache Budget of SLM|20%|40%|60%|80%|Full|
> |-|-|-|-|-|-|
> |Accuracy (BBH)|0.2873|0.4416|0.4624|0.4642|0.4359|
>
> Furthermore, in an independently deployed scenario, the SLM is not required to generate draft tokens, enabling the adoption of an early layer stopping strategy. This approach significantly reduces both latency and memory overhead. We fix the LLM’s KV cache budget at 20% and progressively reduce the number of layers utilized in the SLM for attention mapping. The results indicate that stopping at layer 20 (83% of total layers) incurs no loss in accuracy. Furthermore, even when halting at layer 16 (67%), the accuracy of SmallKV remains superior to that of H2O.
>
> |Stop Layer of SLM|24 (Full)|20|18|16|12|H2O|
> |-|-|-|-|-|-|-|
> |Accuracy (BBH)|0.436|0.436|0.390|0.359|0.335|0.318|
>
> So in practice, the SLM’s actual KV cache usage is calculated as: (1/4.67) × 40%(SLM compression) × (20/24)(early stopping) ≈  7.14% of the full LLM KV cache.
>
> **Weakness 2:
> While some ablations are provided, more sensitivity experiment like varying sizes of SLM is missing**
>
> **R**: We apologize if the connection between this experiment and the requested sensitivity analysis was not immediately clear. We hope the following clarification confirms that the impact of SLM size variation is examined and we will highlight the sensitivity experiment in the paper.
>
> The impact of SLM scaling is investigated in Section 5.5 (Impact of SLM Scaling). In this section, we present a dedicated experiment where we systematically vary the size of the SLM. We evaluate SmallKV using a range of SLMs: Qwen2.5 [0.5B, 1.5B, 3B, and 7B], with Qwen2.5-14B serving as the LLM across all configurations.
>
> The results in Figure 7, provide a sensitivity analysis of SmallKV performance with respect to SLM scale. Our findings indicate that while SLM size has a relatively modest effect under higher KV cache budgets, it becomes significantly more influential under extremely low KV cache conditions—an important observation highlighted in the manuscript.
>
> We hope this response has adequately addressed your concerns. We look forward to the opportunity for additional discussion during the review process.

---

### Official Review · Reviewer_7fen · 2025-07-03

**Clarity:** 3
**Significance:** 3
**Originality:** 3
**Rating:** 4
**Confidence:** 3

**Summary:**

This paper introduces SmallKV, a small-model-assisted compensation framework for KV cache compression in LLMs, addressing saliency shift and marginal token over-compression issues. It leverages attention similarity between small and large models to preserve critical information and approximate marginal tokens during decoding. Experiments on GSM8K, BBH, MT-Bench, and LongBench demonstrate that SmallKV maintains high accuracy even with 5% KV cache budget, outperforming baselines like H2O and PyramidInfer. Efficiency evaluations show 1.75–2.56× throughput gains while remaining compatible with Flash Attention and speculative decoding. The method is robust across model scales (7B–72B) and series (Qwen, LLaMA), offering a practical solution for resource-constrained LLM inference.

**Questions:**

- How robust is the SLM compensation in multi-turn dialogues where attention patterns shift rapidly? Could you add dynamic decoding tests to verify this?
- The current fixed definition of marginal tokens may not suit all tasks (e.g., code vs. creative writing). Would you consider task-adaptive criteria for better flexibility？
- Your longest test is 16K tokens, but real-world applications use 128K+. How does error accumulation in SLM approximations affect output quality at such scales?

The final score depends on feedback of the rebuttal, and I would be happy to improve my final score.

**Ethical Concerns:**

["NO or VERY MINOR ethics concerns only"]

**Final Justification:**

The paper is clearly written, the experiments are thorough, and the rebuttal has fully addressed my concerns. Overall, this is a highly practical contribution, and I support its acceptance.

**Limitations:**

yes

**Paper Formatting Concerns:**

The first reference seems to be written in a way that can be simplified a bit.

**Quality:**

4

**Strengths And Weaknesses:**

Strengths:
- The paper is well-written and easy to follow, clearly explaining the motivation, methodology, and results without unnecessary jargon.
- Extensive experiments across multiple benchmarks and model sizes (7B–72B) convincingly demonstrate SmallKV’s superior performance, especially under aggressive KV cache budgets.
- The proposed small-model-assisted compensation effectively addresses saliency shift and marginal token over-compression, achieving 1.75–2.56× throughput gains while maintaining accuracy, significantly outperforming prior eviction-based methods.

Weaknesses:
- While the high similarity between SLM and LLM attention matrices is empirically validated, the paper lacks rigorous theoretical analysis to explain why this consistency exists across different model scales, potentially limiting the generalizability of the approach.
- Although the authors mention that the memory and computational costs of the SLM can be mitigated by sharing resources with speculative decoding, they do not quantify the actual overhead when SmallKV is deployed independently (e.g., increased latency or resource consumption), which may require further optimization in practice.
- The paper demonstrates that SLM scaling significantly impacts performance under extremely low KV cache budgets (e.g., 5%), but it does not explore the feasibility of even smaller models (e.g., 0.1B), which could restrict its applicability on ultra-edge devices.

---

> ### Author Rebuttal · Authors · 2025-07-30
>
> Thank you for your detailed review and valuable feedback! Below, we address the comments you raised:
>
> **Q1: How robust is the SLM compensation in multi-turn dialogues where attention patterns shift rapidly? Could you add dynamic decoding tests to verify this?**
>
> **R**: In Figure 5, we present the results of two-round dialogues over MT-Bench. The experimental results show that SmallKV significantly outperforms other baselines. To further assess the robustness of SmallKV's compensation strategy in more complex dialogue scenarios, we constructed a five-round dialogue test based on BBH test. Each round consists of questions from different subtasks.
>
> The robustness depends on whether the attention mapping relationship between the small model (SLM) and the big model (LLM) changes during multiple rounds of prefilling and decoding. We first verify the SmallKV’s stable attention mapping on Qwen2-7B & Qwen2-0.5B. During the evaluation, we record the cosine similarity of attention patterns in the mapping dictionary constructed by SmallKV, and compare it against both the globally optimal mapping (i.e., the maximum achievable similarity) and randomly generated mappings. As summarized in the table below, the similarity achieved by SmallKV consistently stays close to the upper bound derived from the global perspective across all dialogue turns. This finding indicates that the attention mapping established by SmallKV remains robust and reliable even in extended multi-turn conversation scenarios.
>
> | |turn1|turn2|turn3|turn4|turn5|
> |-|-|-|-|-|-|
> |global|0.822|0.812|0.799|0.796|0.792|
> |SmallKV|0.822|0.809|0.792|0.791|0.787|
> |random|0.516|0.509|0.510|0.514|0.512|
>
> We evaluate the accuracy on the five-turn dialogue BBH test. When the KV cache budget is set to 20%, SmallKV achieves an accuracy of 0.373, significantly outperforming H2O, which achieves 0.308.
>
> **Q2: The current fixed definition of marginal tokens may not suit all tasks (e.g., code vs. creative writing). Would you consider task-adaptive criteria for better flexibility?**
>
> **R**: Thanks for your insightful suggestions. We agree with you that using a fixed threshold for marginal tokens may not yield optimal results. To ensure fair comparability, in our evaluation, we adopted the same experimental parameters setting as other baselines (heavy : [recent + compensation] = 1:1). Notice that the key advantage of SmallKV lies in the small model’s ability to adaptively identify important tokens, we can dynamically define the proportions of critical tokens and marginal tokens based on the recall rate of attention scores from SLM, as illustrated in Figure 2. For example, we define the critical tokens as those accounting for the top 80% of cumulative attention scores, while marginal tokens are set to the subsequent 15% (i.e., from 80% to 95% of the cumulative sum). This approach enables effective tailoring of these proportions to different tasks.
>
> **Q3：Your longest test is 16K tokens, but real-world applications use 128K+. How does error accumulation in SLM approximations affect output quality at such scales?**
>
> **R**: Given that Qwen2 and Qwen2.5 support a maximum context length of 32K, we evaluated the impact of SLM-based approximations on output quality in extremely long contexts by padding the BBH test context with irrelevant text to reach 32K (long context) compared with original BBH test (short context). The results for Qwen2-0.5B and Qwen2-7B across different KV cache budgets are presented in the table below. Under long context scenarios (32K tokens), the accuracy of SmallKV does not degrade.
>
> |KV Cache Budget|32K BBH (long context)|Original BBH (short context)|
> |-|-|-|
> | 0.3 | 0.516| 0.466 |
> | 0.2 | 0.472 | 0.436 |
> | 0.1 | 0.343 | 0.28|
>
> In practice, we observe that LLMs exhibit higher attention sparsity and more uniform attention patterns in long contexts—characteristics that benefit both KV cache compression and SmallKV. Consequently, the model’s performance actually improves when incorporating such long contexts.
>
> **Weakness 1: While the high similarity between SLM and LLM attention matrices is empirically validated, the paper lacks rigorous theoretical analysis to explain why this consistency exists across different model scales, potentially limiting the generalizability of the approach.**
>
> **R**: Though it is hard to provide a rigorous theoretical proof, we aim to demonstrate the similarity between SLM and LLM attention patterns across different model scales through extensive experimentation. The results of these tests are summarized in the table below. In addition to the model combinations presented in the evaluation section, we have further extended our experiments to include Qwen3 series, as well as cross-series combination between Qwen3 and Qwen2. (Note1: Because of differences of tokenizers in different series models, we compute pairwise longest common subsequences to match tokens and retain only the matched tokens for calculating cosine similarity.) (Note2: Due to the presence of attention sinks, we excluded the initial few tokens during similarity calculations to prevent artificially inflated similarity scores. (e.g. ~0.99), ensuring meaningful and comparable results.) Our results show that, although some variation exists, this similarity is consistently present across different datasets.
>
> |SLM|Qwen2-0.5B|Qwen2-0.5B|Qwen2-0.5B|Qwen3-0.6B|Qwen2.5-0.5B|Qwen2.5-1.5B|Qwen2.5-3B|Qwen2.5-7B|LLaMA3.2-1B|
> |-|-|-|-|-|-|-|-|-|-|
> |LLM|Qwen2-7B|Qwen2-7B|Qwen2-7B|Qwen3-8B|Qwen2.5-14B|Qwen2.5-14B|Qwen2.5-14B|Qwen2.5-14B|Qwen2-7B|
> |Benchmark|BBH|GSM8K|MMLU|BBH|BBH|BBH|BBH|BBH|BBH|
> |Similarity|0.85|0.85|0.98|0.83|0.83|0.83|0.85|0.86|0.56|
>
> **Weakness 2: Although the authors mention that the memory and computational costs of the SLM can be mitigated by sharing resources with speculative decoding, they do not quantify the actual overhead when SmallKV is deployed independently (e.g., increased latency or resource consumption), which may require further optimization in practice.**
>
> **R**: We quantify the theoretical maximum latency and memory overhead of the model combinations used in main experiments, as shown in the table below. The values 'Ratio' represent the additional overhead introduced by SLM relative to LLM (e.g., for combination Qwen2-0.5B & 7B, the KV cache of SLM is 1/4.67 that of the LLM).
>
> |Ratio between SLM and LLM|Qwen2-0.5B & Qwen2-7B|Qwen2.5-0.5B & Qwen2.5-14B|Qwen2-7B & Qwen2-72B|LLAMA 3.2-1B & LLAMA 3.1-8B|
> |--|--|--|--|--|
> |Latency|1/3.57|1/6.52|1/4.47|1/3.19|
> |KV cache|1/4.67|1/16.0|1/5.71|1/4.0|
>
> We highlight the practicality of SmallKV, emphasizing how the computational overhead of the SLM can be effectively mitigated through deployment-level optimizations. To achieve this, we propose three key strategies: (1) integration with speculative decoding (SP); (2) compression of the KV cache for the SLM; and (3) layer skipping (i.e., early stopping) during the attention mapping process.
>
> Regarding latency, while the forward pass of the SLM during decoding incurs some overhead, this is mitigated by the acceleration provided through KV cache compression. As demonstrated in Section 5.3 Table 2 of the evaluation, SmallKV achieves lower latency compared to the baseline model. Additionally, the integration of SP can eliminate the forward pass of the SLM as a significant latency bottleneck. Current SP algorithms yield 3-5x speedups, further enhancing performance.
>
> Regarding memory consumption, the high attention sparsity inherent in SLM enables near-lossless reduction in memory usage through SLM KV cache compression. As shown in the table below, we fix the KV cache budget of the LLM (Qwen2-7B) at 20% and increase the compression ratio of the SLM (Qwen2-0.5B) to evaluate the LLM’s accuracy on BBH (SmallKV ACC = 0.436 at 20%). Under the same memory constraint (for SmallKV we count both SLM and LLM cache), the performance of SmallKV (ACC = 0.462 at 40% for SLM and 20% for LLM) outperforms H2O (ACC = 0.38 at 30%). Note that while SLM compression may slightly impair saliency shift compensation, the trade-off remains favorable in terms of overall efficiency.
>
> |KV Cache Budget of SLM|20%|40%|60%|80%|Full|
> |-|-|-|-|-|-|
> |Accuracy (BBH)|0.2873|0.4416|0.4624|0.4642|0.4359|
>
> Furthermore, in an independently deployed scenario, the SLM is not required to generate draft tokens, enabling the adoption of an early layer stopping strategy. This approach significantly reduces both latency and memory overhead. We fix the LLM’s KV cache budget at 20% and progressively reduce the number of layers utilized in the SLM for attention mapping. The results indicate that stopping at layer 20 (83% of total layers) incurs no loss in accuracy (0.436). Furthermore, even when halting at layer 16 (67%), the accuracy of SmallKV remains superior to that of H2O.
>
> |Stop Layer of SLM|24 (Full)|20|18|16|12|H2O|
> |-|-|-|-|-|-|-|
> |Accuracy (BBH)|0.436|0.436|0.390|0.359|0.335|0.318|
>
> **Weakness 3: The paper demonstrates that SLM scaling significantly impacts performance under extremely low KV cache budgets (e.g., 5%), but it does not explore the feasibility of even smaller models (e.g., 0.1B), which could restrict its applicability on ultra-edge devices.**
>
> **R**: The main limitation hindering the scaling of the SLM to smaller sizes (e.g., 0.1B parameters) is the absence of such compact variants within major open-source model families. To address this challenge, we plan to explore techniques such as model distillation to train smaller versions of the SLM. This will enable us to better meet the deployment requirements of SmallKV on ultra-edge devices.
>
> **Paper Formatting Concerns**
>
> **R**: Thank you for your suggestion. Revision: Phi Team et al. Phi-3 technical report: A highly capable language model locally on your phone. arXiv preprint arXiv:2404.14219, 2024.
>
> We hope this response has adequately addressed your concerns. We look forward to the opportunity for additional discussion during the review process.

---

> > ### Comment · Reviewer_7fen · 2025-08-05
> > **Official comment of 7fen**
> >
> > Thank you for your response, which has largely addressed my concerns. SmallKV indeed represents a very practical contribution. I will maintain my current score for now and may consider increasing it based on further discussions with the other reviewers.

---

> > > ### Author Response · Authors · 2025-08-05
> > >
> > > Thank you very much for your recognition of our work. We are delighted to receive your feedback. If you have any further questions, please feel free to comment at any time, and we will promptly address your concerns.

---

### Comment · Area_Chair_Wxui · 2025-08-02
**Please Engage with Authors’ Responses During Rebuttal**

Dear Reviewers,

As we approach the rebuttal phase, I want to take a moment to remind you of the importance of carefully reviewing the authors’ responses. The rebuttal process is a valuable opportunity for authors to clarify misunderstandings, address concerns, and provide additional evidence or analysis to support their work.

When reviewing the authors’ responses, please:

1. Read their replies thoroughly to ensure you understand their points.
2. Assess whether your concerns have been adequately addressed.

Your engagement in this process helps ensure a fair and constructive review outcome. Thank you for your time and dedication to maintaining the high standards of NeurIPS. Your thoughtful participation in this phase is greatly appreciated!

Best regards,

Area Chair

---

### Decision · Program_Chairs · 2025-09-17

**Decision:**

Accept (spotlight)

**Comment:**

## Summary
The paper proposes SmallKV, a novel framework for KV cache compression in large language models (LLMs) that leverages a small language model (SLM) to address saliency shift and marginal token over-compression issues. By utilizing attention similarity between SLMs and LLMs, SmallKV achieves efficient compression while maintaining high accuracy across various benchmarks and model scales (7B–72B). The method demonstrates significant throughput improvements (1.75–2.56×) and compatibility with existing techniques like Flash Attention and speculative decoding.

## Strengths
1. Reviewers unanimously praise the paper’s clarity and accessibility, noting its well-written structure and clear explanation of motivation, methodology, and results.
2. The innovative use of an SLM to guide KV cache compression, particularly in addressing saliency shift and marginal token over-compression, is highlighted as a novel and impactful contribution by all reviewers.
3. Extensive experiments across diverse benchmarks (GSM8K, BBH, MT-Bench, LongBench) and model scales demonstrate SmallKV’s robustness and superior performance compared to baselines like H2O and PyramidInfer, especially under aggressive KV cache budgets.
4. The Jaccard similarity matching of SLM-LLM attention matrices and dynamic eviction strategies further enhance the method’s effectiveness. Additionally, SmallKV’s compatibility with existing optimization techniques makes it practical for real-world deployment.

## Weaknesses
1. Reviewers raise concerns about the lack of rigorous theoretical analysis to explain the observed high similarity in attention patterns between SLMs and LLMs, which may limit the method’s generalizability across diverse models and tasks.
2. The computational and memory overhead of using an SLM, particularly when deployed independently, is not adequately quantified, raising practical concerns about latency and resource consumption.
3. The evaluation is limited to simpler tasks (e.g., GSM8K), and its applicability to more complex problems, such as those in AMIE or AMC, remains unclear (Reviewer rx3H). Additionally, the absence of experiments with smaller SLMs (e.g., 0.1B) or negative comparisons across different model series restricts insights into SmallKV’s flexibility on ultra-edge devices or diverse architectures.
4. Finally, comparisons against dated baselines like H2O rather than more recent state-of-the-art methods weaken the paper’s competitive claims.

**After rebuttal**, reviewers appreciate that their concerns are addressed.

Consider the provided insights of this work are interesting, and the results are remarkable, I recommend accepting this paper as a spotlight.